# Individual Arbitrariness and Group Fairness

**Carol Xuan Long**[‡]**, Hsiang Hsu**[*†]**, Wael Alghamdi**[*‡]**, Flavio P. Calmon**[‡]

## Abstract

Machine learning tasks may admit multiple competing models that achieve similar performance yet produce arbitrary outputs for individual samples—a phenomenon known as predictive multiplicity. We demonstrate that fairness interventions in machine learning optimized solely for group fairness and accuracy can exacerbate predictive multiplicity. Consequently, state-of-the-art fairness interventions can mask high predictive multiplicity behind favorable group fairness and accuracy metrics. We argue that a third axis of "arbitrariness" should be considered when deploying models to aid decision-making in applications of individual-level impact. To address this challenge, we propose an ensemble algorithm applicable to any fairness intervention that provably ensures more consistent predictions.

## 1 Introduction

Non-arbitrariness is an important facet of non-discriminatory decision-making. Substantial arbitrariness exists in the training and selection of machine learning (ML) models. By simply varying hyperparameters of the training process (e.g., random seeds in model training), we can produce models with arbitrary outputs on individual input samples [9, 13, 26, 35]. The phenomenon where distinct models exhibit similar accuracy but arbitrary individual predictions is called *predictive multiplicity*[1] [35]. The arbitrary variation of outputs due to unjustified choices made during training can disparately impact individual samples, i.e., predictive multiplicity is not equally distributed across inputs of a model. When deployed in high-stakes domains (e.g., medicine, education, resume screening), the arbitrariness in the ML pipeline may target and cause systemic harm to specific individuals by excluding them from favorable outcomes [7, 15, 40].

Popular fairness metrics in the ML literature do not explicitly capture non-arbitrariness. A widely recognized notion of non-discrimination is *group fairness*. Group fairness is quantified in terms of, for example, statistical parity [18], equal opportunity, equalized odds [23], and variations such as multi-accuracy [30] and multi-calibration [24]. Broadly speaking, methods that control for group fairness aim to guarantee comparable performance of a model across population groups in the data. The pursuit of group fairness has led to hundreds of fairness interventions that seek to control for performance disparities while preserving accuracy [25].

The central question we tackle in this paper is: Do models corrected for group fairness exhibit less arbitrariness in their outputs? We answer this question in the *negative*. We demonstrate that state-of-the-art fairness interventions may improve group fairness metrics at the expense of exacerbating arbitrariness. The harm is silent: the increase in arbitrariness is masked by favorable group fairness and accuracy metrics. Our results show that arbitrariness lies beyond the fairness-accuracy frontier:

---

[*]equal contributions.

[†]Prepared prior to employment at JPMorgan Chase & Co.. Email: `hsiang.hsu@jpmchase.com`.

[‡]John A. Paulson School of Engineering and Applied Sciences, Harvard University, Boston, MA 02134. Emails: `carol_long@g.harvard.edu, alghamdi@g.harvard.edu, flavio@seas.harvard.edu`.

[1]In the rest of the paper, we informally use the terms "arbitrariness" and "predictive multiplicity" exchangeably to refer to the phenomenon of inconsistent predictions caused by the randomness in model training.

37th Conference on Neural Information Processing Systems (NeurIPS 2023).

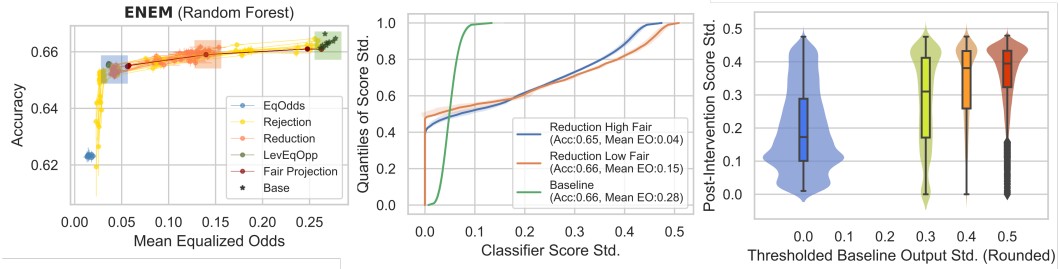

**Figure 1:** Accuracy-fairness frontier does not reveal arbitrariness in competing models. **Left**: Fairness-Accuracy frontier of baseline and fair models corrected by 5 fairness interventions; point clouds generated by different random seed choices. **Middle**: The cumulative distribution functions (CDF) of per-sample score std. across classifiers at different intervention levels (see Definition 3). For each sample, std. is measured across competing scores produced by classifiers initialized with different random seeds. A *wider* CDF indicates *more* disparity of the impact of arbitrariness on different individuals. **Right**: The distribution of score std. relative to the thresholded baseline model. Removing samples that receive very low score std. both from thresholded baseline and fair classifiers, the largest group (blue area) in this violin plot are those individuals for which std. increases from 0 to a large positive value (median around 0.15). Hence, significant arbitrariness is introduced by the fairness intervention, in addition to and separate from the effects of thresholding the baseline.

predictive multiplicity should be accounted for *in addition* to usual group-fairness and accuracy metrics during model development.

Figure 1 illustrates how fairness interventions can increase predictive multiplicity. Here, state-of-the-art fairness interventions are applied[4] to a baseline random forest classifier to ensure group fairness (mean equalized odds [23], see Definition 6) in a student performance binary prediction task. We produce multiple baseline classifiers by varying the random seed used to initialize the training algorithm. Each baseline classifier achieves comparable accuracy and fairness violation. They also mostly agree in their predictions: for each input sample, the standard deviation of output scores across classifiers is small (see Definition 3). After applying a fairness intervention to each randomly initialized baseline classifier, we consistently reduce group fairness violations at a small accuracy cost, as expected. However, predictive multiplicity changes significantly post-intervention: for roughly half of the students, predictions are consistent across seeds, whereas for 20% of the students, predictions are comparable to a coin flip. For the latter group, the classifier output depends on the choice of a random seed instead of any specific input feature. The increase in predictive multiplicity is masked by the fairness-accuracy curve, does not impact all samples equally, and is consistent across datasets and learning tasks.

At first, the increase in predictive multiplicity may seem counter-intuitive: adding fairness constraints to a learning task should reduce the solution space, leading to less disagreement across similarly-performing classifiers relative to an unconstrained baseline. We demonstrate that, in general, this is not the case. For a given hypothesis class, the non-convex nature of group fairness constraints can in fact *increase* the number of feasible classifiers at a given fairness and accuracy level. We show that this phenomenon occurs even in the simple case where the hypothesis space is comprised of threshold classifiers over one-dimensional input features, and the optimal baseline classifier is unique. To address this challenge, we demonstrate – both theoretically and through experiments – that ensembling classifiers is an effective strategy to counteract this multiplicity increase.

The main contributions of this work include[5]:

1. We demonstrate that the usual "fairness-accuracy" curves can systematically mask an increase of predictive multiplicity. Notably, applying state-of-the-art fairness interventions can incur higher arbitrariness in the ML pipeline.

2. We show that multiplicity can be arbitrarily high even if group fairness and accuracy are controlled, when we do not have perfect classifiers. Hence, fairness interventions optimized solely for fairness and accuracy cannot, in general, control predictive multiplicity. We also provide examples of why fairness constraints may exacerbate arbitrariness.

---

[4]See Section 5 for a detailed description of the experiment and dataset.

[5]Proofs and additional experiments are included in the Appendices.

3. We propose an ensemble algorithm that reduces multiplicity while maintaining fairness and accuracy. We derive convergence rate results to show that the probability of models disagreeing drops exponentially as more models are added to the ensemble.

4. We demonstrate the multiplicity phenomena and benchmark our ensemble method through comprehensive experiments using state-of-the-art fairness interventions across real-world datasets.

## 1.1 Related Works

**Multiplicity, its implications, and promises.** Recent works have investigated various factors that give rise to multiplicity. D'Amour et al. [17] studied how under-specification presents challenges to the credibility of modern machine learning algorithms. More precisely, under-specified optimization problems in machine learning admit a plethora of models that all attain similar performance, and which model to deploy in practice may ultimately depend on arbitrary choices of the randomization made during training procedure [3]. The arbitrariness of the model could potentially harm the reproducibility of model predictions [5], and hence the credibility of the conclusion made thereof.

Creel and Hellman [15] thoroughly explored the notion of arbitrariness in machine learning and discuss how high-multiplicity predictions can lead to systematized discrimination in society through "algorithmic leviathans." Multiplicity in prediction and classification can also have beneficial effects. Black et al. [7], Semenova et al. [39], and Fisher et al. [20] view multiplicity of equally-performing models as an opportunity to optimize for additional criteria such as generalizability, interpretability, and fairness. Coston et al. [14] develop a framework to search over the models in the Rashomon set for a better operation point on the accuracy-fairness frontier. However, they do not discuss the potential predictive multiplicity cost of existing fairness interventions nor propose algorithms to reduce this cost.

The work most similar to ours is [13]. Cooper et al. [13] consider the problem of predictive multiplicity as a result of using different splits of the training data. Therein, they quantify predictive multiplicity by prediction variance, and they propose a bagging strategy [8] to combine models. Our work considers a different problem where predictive multiplicity is exacerbated by group-fairness interventions. Our work is also different from Cooper et al. [13] as we fix the dataset when training models and consider multiplicity due to randomness used during training. In this sense, our ensemble algorithm is actually a voting ensemble [41] (see Section 4); see also ensembling and reconciliation strategies proposed by Black et al. [6] and Roth et al. [38] that aim to create more consistent predictions among competing models. To the best of the authors' knowledge, we are the first to measure and report the arbitrariness cost of fairness interventions.

**Hidden costs of randomized algorithms.** Recent works [21, 31, 32] examine the potential detrimental consequences of randomization in the ML pipeline. In their empirical study, Ganesh et al.[21] observe that group fairness metrics exhibit high variance across models at different training epochs of Stochastic Gradient Descent (SGD). The authors point out that random data reshuffling in SGD makes empirical evaluation of fairness (on a test set) unreliable, and they attribute this phenomenon to the volatility of predictions in minority groups. Importantly, they do not incorporate fairness interventions in their experiments. In contrast, we apply fairness interventions to baseline models. Specifically, we examine the variance in predictions among models with similar fairness and accuracy performances. In addition to the observations made by Ganesh et al., our theoretically-grounded study reveals the different paths that lead to group-fairness, i.e., that arbitrariness can be an unwanted byproduct of imposing fairness constraints. Krco et al. [31] empirically study if fairness interventions reduce bias equally across groups, and examine whether affected groups overlap across different fairness interventions. In contrast, our work examines the *multiplicity cost* of group fairness and its tension with individual-level prediction consistency, rather than the *fairness cost* of randomness in the ML pipeline. Another work on the hidden cost of randomized algorithms is given by Kulynych et al. [32], who report that well-known differentially-private training mechanisms can exacerbate predictive multiplicity.

In an early work [34], Lipton et al. indirectly points to the potential arbitrary decision on individuals as a result of imposing group fairness constraints. They give an illustrative example using synthetic hiring data to show that a fair model resorts to using irrelevant attribute (hair length) to make hiring decision in order to achieve near-equal hiring rate for men and women.

## 2 Problem Formulation and Related Works

We explain the setup and relevant definitions in this section.

**Prediction tasks.** We consider a binary classification setting with training examples being triplets $(\boldsymbol{X}, S, Y)$ with joint distribution $P_{\boldsymbol{X},S,Y}$. Here, $\boldsymbol{X}$ is an $\mathbb{R}^d$-valued feature vector, $S$ is a discrete random variable supported on $[K] \triangleq \{1, \cdots, K\}$ representing $K$ (potentially overlapping) group memberships, and $Y$ is a binary (i.e., $\{0, 1\}$-valued) random variable denoting class membership.[6] We consider probabilistic classifiers in a hypothesis space $\mathcal{H}$, where each $h \in \mathcal{H}$ is a mapping $h : \mathbb{R}^d \to [0, 1]$. Each value of a classifier $h(\boldsymbol{x})$ aims to approximate $P_{Y|\boldsymbol{X}=\boldsymbol{x}}(1)$. The predicted labels $\widehat{y}$ can be obtained by thresholding the scores, e.g., $\widehat{y} = \mathbb{1}\{h(\boldsymbol{x}) \geq 0.5\}$, where $\mathbb{1}\{\cdot\}$ is the indicator function. Finally, we denote by $\boldsymbol{\Delta}_c$ the probability simplex over $c$ dimensions.

**Randomized training procedures and the Rashomon set.** We assume access to the following:

1. a training dataset of $n$ i.i.d samples $\mathcal{D} \triangleq \{(\boldsymbol{x}_i, s_i, y_i)\}_{i=1}^n$ drawn from $P_{\boldsymbol{X},S,Y}$;
2. a randomized training procedure $\mathcal{T}$; and
3. an induced distribution $\mathcal{T}(\mathcal{D})$ on the hypothesis class of predictors $\mathcal{H}$.

We denote a sampled classifier by $h \sim \mathcal{T}(\mathcal{D})$, which can be sampled, for example, using different random seeds at the beginning of the execution of procedure $\mathcal{T}$ on $\mathcal{D}$. For concreteness, the above data may for example correspond to the following practical setting.

**Example 1.** The dataset $\mathcal{D}$ can be comprised of resumes of individuals applying for a job, and the training procedure $\mathcal{T}$ is an algorithm to predict whether to extend an interview opportunity for an applicant. For example, $\mathcal{T}$ can be a neural network with unspecified hyperparameters (e.g., random seed that needs to be chosen at the outset); alternatively, $\mathcal{T}$ can be the same pre-trained neural network composed with a fairness intervention method. The classifiers considered will be the last layer of the neural network (or the classifier after fairness enhancement), which will belong to a hypothesis class $\mathcal{H}$ determined by the chosen neural network architecture. By varying the random seed, say, $m$ times, we would obtain *independent* classifiers, denoted by $h_1, \cdots, h_m \overset{i.i.d.}{\sim} \mathcal{T}(\mathcal{D})$. ♦

We are interested in detecting whether competing classifiers (i.e., deployed for, and performing similarly in the same prediction task) have conflicting predictions non-uniformly across individuals. Next, we define the set of competing models obtained from the randomized training procedure $\mathcal{T}$.

For a loss function $\ell : [0, 1] \times \{0, 1\} \to \mathbb{R}^+$, finite dataset $\mathcal{D} \subset \mathbb{R}^d \times [K] \times \{0, 1\}$, and classifier $h : \mathbb{R}^d \to [0, 1]$, we let the empirical loss incurred by $h$ on $\mathcal{D}$ be denoted by $\ell(h; \mathcal{D}) \triangleq |\mathcal{D}|^{-1} \sum_{(\boldsymbol{x},s,y) \in \mathcal{D}} \ell(h(\boldsymbol{x}), y)$. The (*empirical*) *Rashomon set* ($\epsilon$-level set) of competing models is defined as the set of models with loss lower than $\epsilon$ [26], i.e., $\mathcal{R}(\mathcal{H}, \mathcal{D}, \epsilon) \triangleq \{h \in \mathcal{H} : \ell(h; \mathcal{D}) \leq \epsilon\}$. We extend the definition of the Rashomon set to take into consideration the effect of the randomized algorithm $\mathcal{T}$, as follows.

**Definition 1** (Empirical Rashomon Set of Randomized Training Procedure)**.** Fix a finite dataset $\mathcal{D}$, a hypothesis class $\mathcal{H}$, and a randomized training procedure $\mathcal{T}$ inducing the distribution $\mathcal{T}(\mathcal{D})$ on $\mathcal{H}$. Given a loss function $\ell : [0, 1] \times \{0, 1\} \to \mathbb{R}^+$ and a parameter $\epsilon > 0$, we define the *empirical Rashomon set with $m$ models induced by $\mathcal{T}$* as the collection of $m$ classifiers independently sampled from $\mathcal{T}(\mathcal{D})$ and having empirical loss less than $\epsilon$:

$$\widehat{\mathcal{R}}_m(\mathcal{T}, \mathcal{D}, \epsilon) \triangleq \left\{ h_1, \cdots, h_m \in \mathcal{H} : h_1, \cdots, h_m \overset{i.i.d.}{\sim} \mathcal{T}(\mathcal{D}) \text{ and } \ell(h_j; \mathcal{D}) \leq \epsilon \, \forall j \in [m] \right\}. \quad (1)$$

Here, $\epsilon$ is an approximation parameter that determines the size of the set. The set $\widehat{\mathcal{R}}_m(\mathcal{T}, \mathcal{D}, \epsilon)$ can be viewed as an approximation of the *Rashomon set* of "good" models [9, 35, 40], and indeed we have the inclusion $\widehat{\mathcal{R}}_m(\mathcal{T}, \mathcal{D}, \epsilon) \subset \mathcal{R}(\mathcal{H}, \mathcal{D}, \epsilon)$ where $\mathcal{H} = \text{supp}(\mathcal{T}(\mathcal{D}))$. Note that the set $\widehat{\mathcal{R}}_m(\mathcal{T}, \mathcal{D}, \epsilon)$ is itself *random* even for a fixed dataset $\mathcal{D}$, where the source of randomness is coming from the distribution $\mathcal{T}(\mathcal{D})$. In the sequel, we omit the arguments of $\widehat{\mathcal{R}}_m(\mathcal{T}, \mathcal{D}, \epsilon)$ when they are clearly implied from context.

---

[6]We note that our setup can be readily extended to multi-class prediction.

There are various metrics to quantify predictive multiplicity across models in $\widehat{\mathcal{R}}_m$ by either considering their output scores [40] or thresholded predictions [35]. e focus on two metrics: 1) *ambiguity* for evaluating the predictive multiplicity of thresholded predictions, and 2) *cumulative distribution function (CDF) of standard deviation (std.) of output scores* when model outputs are in the interval $[0, 1]$ (interpreted as the probability of the positive class). Those two metrics are defined as follows.

**Definition 2** (Ambiguity [35]). Fix a dataset $\mathcal{D} = \{(\boldsymbol{x}_i, s_i, y_i)\}_{i \in [n]} \subset \mathbb{R}^d \times [K] \times \{0, 1\}$ and a finite set of models $\mathcal{R} \subset \mathcal{H}$. Let $f(r) \triangleq \mathbb{1}\{r \geq 0.5\}$ be the thresholding function. The *ambiguity* of a dataset over the set of models $\mathcal{R}$ is the proportion of points in the dataset that can be assigned a conflicting prediction by a competing classifier within $\mathcal{R}$:

$$\alpha\left(\mathcal{D}, \mathcal{R}\right) \triangleq \frac{1}{|\mathcal{D}|} \sum_{i \in [n]} \max_{h, h' \in \mathcal{R}} \mathbb{1}\left\{f(h(\boldsymbol{x}_i)) \neq f(h'(\boldsymbol{x}_i))\right\}. \tag{2}$$

To define the CDF of std. of scores, we first delineate what we mean by empirical std. of scores.

**Definition 3** (Std. of Scores). Fix a finite set of models $\mathcal{R} = \{h_j\}_{j \in [m]} \subset \mathcal{H}$. The empirical standard deviation (std.) of scores for a sample $\boldsymbol{x} \in \mathbb{R}^d$ relative to $\mathcal{R}$ is defined by

$$s(\boldsymbol{x}, \mathcal{R}) \triangleq \sqrt{\frac{1}{m - 1} \sum_{j \in [m]} (h_j(\boldsymbol{x}) - \bar{\mu}_{\boldsymbol{x}})^2}, \tag{3}$$

where $\bar{\mu}_{\boldsymbol{x}} \triangleq \frac{1}{m} \sum_{j \in [m]} h_j(\boldsymbol{x})$ denotes the empirical mean (over $\mathcal{R}$) of the scores.

Further, to understand the std. of scores of the population, we consider the empirical cumulative distribution function of the std. of the scores, defined as follows.

**Definition 4** (Quantiles of std. of Scores). Fix a dataset $\mathcal{D} = \{(\boldsymbol{x}_i, s_i, y_i)\}_{i \in [n]} \subset \mathbb{R}^d \times [K] \times \{0, 1\}$ and a finite set of models $\mathcal{R} \subset \mathcal{H}$. We define the empirical cumulative distribution function of the std. of the scores by (where $s(\boldsymbol{x}, \mathcal{R})$ is the empirical std. as in Definition 3)

$$\widehat{F}_{\mathcal{D}, \mathcal{R}}(t) \triangleq \frac{1}{|\mathcal{D}|} \sum_{i \in [n]} \mathbb{1}\left\{s(\boldsymbol{x}_i, \mathcal{R}) \leq t\right\}. \tag{4}$$

**Example 2.** Consider a resume screening task where the algorithm decides whether to extend an interview opportunity. If $\widehat{F}_{\mathcal{D}, \mathcal{R}}(0.5) = 90\%$, then for 10% of the individuals in the dataset, the predictions produced by the competing models are arbitrary and conflicting: regardless of the mean scores, with an std. of at least 0.5, there would exist models with scores falling above and below the one-half threshold, so the thresholded output can be both 0 (no interview) and 1 (offer interview). ♦

**A note on related metrics.** An alternative measurement of score variation is *Viable Prediction Range* as defined in [40], which measures the difference in max and min scores among competing models on each individual. For thresholded scores, the original definition of *Ambiguity* [35] considers the proportion of a flip in prediction with respect to a baseline model (from empirical risk minimization with fixed hyperparameters and randomness). Since we consider randomized training procedures with no clear baseline model, the definition for ambiguity above is a variation of the original.

**Group fairness.** We consider three group fairness definitions for classification tasks—statistical parity (SP), equalized odds (EO), and overall accuracy equality (OAE) [11, 18, 23, 37]. OAE and Mean Equalized Odds (MEO) are defined below as they are used in the next sections, and we refer the reader to Appendix D for the remaining definitions.

**Definition 5** (Overall Accuracy Equality, OAE). Let $\widehat{Y}$ be the predicted label obtained, e.g., from thresholding the scores of a classifier $h : \mathbb{R}^d \to [0, 1]$. The predictor $\widehat{Y}$ satisfies overall accuracy equality (OAE) if its accuracy is independent of the group attribute: for all groups $s, s' \in [K]$,

$$\Pr(\widehat{Y} = Y \mid S = s) = \Pr(\widehat{Y} = Y \mid S = s'). \tag{5}$$

For binary classification, SP boils down to requiring the average predictions to be equal across groups, while EO requires true positive rates (TPR) and false positive rates (FPR) to be calibrated. In this paper, we consider mean equalized odds (MEO): the average of absolute difference in FPR and TPR for binary groups $S \in \{0, 1\}$. We consider binary group since this is the setup for most fairness intervention methods.

**Definition 6** (Mean Equalized Odds, MEO [4, 23]). Let $\widehat{Y}$ be the predicted label, $S \in \{0, 1\}$ denotes binary group membership. Mean Equalized Odds is the average odds difference for binary groups:

$$\text{MEO} \triangleq \frac{1}{2}\left(|\text{TPR}_{S=0} - \text{TPR}_{S=1}| + |\text{FPR}_{S=0} - \text{FPR}_{S=1}|\right), \tag{6}$$

where $\text{TPR}_{S=s} \triangleq \Pr(\widehat{Y} = 1 \mid Y = 1, S = s)$ and $\text{FPR}_{S=s} \triangleq \Pr(\widehat{Y} = 1 \mid Y = 0, S = s)$.

To examine whether current fairness intervention methods lead to an exacerbation of multiplicity, we survey state-of-the-art intervention methods, including Reductions [1], Fair Projection [2], Reject Options [29], and EqOdds [23]. We offer a brief discussion of their mechanism in Appendix D.

## 3 Orthogonality of Fairness and Arbitrariness

We discuss next why arbitrariness is a third axis not captured by fairness and accuracy. Models with similar fairness and accuracy metrics can differ significantly in predictions. Moreover, a set of fair and approximately accurate models can attain maximal predictive multiplicity. We also explore through an example one fundamental reason why adding fairness constraints can lead to more arbitrariness.

**Example 3** (Ambiguity ≠ OAE). Overall Accuracy Equality (OAE, Definition 5) does not capture the ambiguity of model outputs (Definition 2). Consider two hypothetical models that are fair/unfair and exhibit high/low predictive multiplicity in Figure 2. Here, in each panel, the rectangle represents the input feature space and the shaded regions represent the error region of each model.

In the top left panel, both Model 1 and 2 have equal accuracy for both groups, since the proportion of the error regions (red stripes and pink shade) for both groups are the same. Hence, both models are considered group-fair in terms of OAE. However, the error regions of the two models are disjoint. Since ambiguity is measured by the percentage of the samples that receive conflicting predictions from either models, samples from the union of the two error regions contribute to ambiguity. Hence, Model 1 and 2 bear high predictive multiplicity despite being group-fair.

In the lower right panel, Model 1 and 2 attain low fairness and low predictive multiplicity. Both models have higher accuracy for Group 2 than Group 1, so they are both unfair. The error regions completely overlap, which means that the two models are making the same error—ambiguity is 0. ♦

The schematic diagram in Figure 2 shows that predictive multiplicity is not captured by OAE. Indeed, ambiguity of a collection of models is a *global* property (i.e., verified at the collection level), whereas OAE is a *local* property (i.e., verified at the classifier level). Hence, one should not *a priori* expect that a set of competing models each satisfying OAE would necessarily comprise a Rashomon set with favorable ambiguity.

We prove the orthogonality of OAE and Statistical Parity (SP) from ambiguity formally in the proposition below, where we show that it is possible to construct classifiers with very stringent accuracy and perfect fairness constraint, albeit with maximal ambiguity. We determine the Rashomon set using the 0-1 loss:

$$\ell_{0\text{-}1}(h; \mathcal{D}) = \frac{1}{|\mathcal{D}|} \sum_{(\boldsymbol{x}_i, s_i, y_i) \in \mathcal{D}} \mathbb{1}\left\{\widehat{y}_i \neq y_i\right\},$$

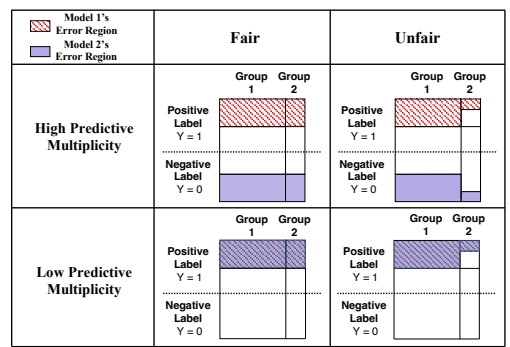

**Figure 2:** Illustration on two models being fair/unfair and exhibit high/low predictive multiplicity through the models' error regions in each of the 4 cases. The metrics for fairness and predictive multiplicity are Overall Accuracy Equality (Definition 5) and ambiguity (Definition 2), respectively.

where $\widehat{y}_i \in \{0, 1\}$ is the class membership of $\boldsymbol{x}_i$ predicted by $h$. We prove the following orthogonality in Appendix A.

**Proposition 3.1** (Orthogonality of OAE/SP and Ambiguity). *Fix any empirical loss value $0 < \epsilon \leq \frac{1}{2}$ and any number of models $m > \frac{1}{\epsilon}$. Then, for some finite dataset $\mathcal{D} \subset \mathbb{R}^d \times [K] \times \{0, 1\}$, there is a realization of the empirical Rashomon set $\widehat{\mathcal{R}}_m = \{h_j\}_{j \in [m]}$ satisfying the following simultaneously:*

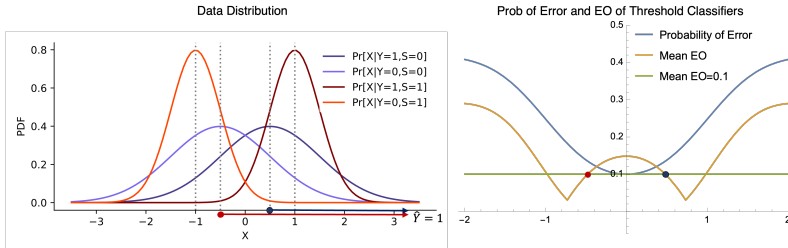

**Figure 3:** Data distribution of a population with two groups used in Example 2 (**Left**). In **Right**, without the Mean EO constraint (6) (green line), there is a unique optimal classifier (with threshold 0) that attains the smallest probability of error (blue line). Adding the Mean EO constraint enlarges the set of optimal threshold classifiers to two classifiers (red and blue dots) with indistinguishable accuracy and fairness levels (**Right**) but different decision regions. We illustrate the decision regions of each classifier as red and blue arrows on the **Left**.

1. *Each $h_j$ has 0-1 loss upper bounded by $\ell_{0\text{-}1}(h_j; \mathcal{D}) \leq \epsilon$;*

2. *Each $h_j$ satisfies OAE perfectly, or each $h_j$ satisfies SP perfectly;*

3. *The collection $\widehat{\mathcal{R}}_m$ has the worst ambiguity, i.e., $\alpha(\mathcal{D}, \widehat{\mathcal{R}}_m) = 100\%$.*

**Remark 1.** For OAE, such Rashomon set $\widehat{\mathcal{R}}_m$ exists for *any* dataset $\mathcal{D}$ satisfying the two conditions:

1. with $n_k$ denoted the number of samples in $\mathcal{D}$ belonging to group $k \in [K]$, the greatest common divisor of the $n_k$ is at least $(m-1)/(m\epsilon - 1)$.

2. if $(\boldsymbol{x}, s, y), (\boldsymbol{x}, s', y') \in \mathcal{D}$ share the same feature vector $\boldsymbol{x}$, then $y = y'$ too.

The requirement that the $n_k$ share a large enough common divisor is used in the proof to guarantee *perfect* OAE. One could relax this requirement at the cost of nonzero OAE violation.

Proposition 3.1 implies that there exists a dataset and competing models for which all samples receive conflicting predictions. Specifically, we can construct a large enough set of competing models ($m > \frac{1}{\epsilon}$) such that 100% of the samples in the dataset can receive conflicting predictions from this set of perfectly fair models with respect to OAE.

In the next example, we demonstrate that, counter-intuitively, adding a fairness constraint can enlarge the set of optimal models, thereby increasing predictive multiplicity. This points to a fundamental reason why adding fairness constraints can lead to more arbitrariness in model decisions.

**Example 4** (Arbitrariness of Threshold Classifiers with Fairness Constraint)**.** Given a data distribution of a population with two groups (Figure 3 **Left**), consider the task of selecting a threshold classifier that predicts the true label. Without fairness considerations, the optimal threshold is 0 – i.e., assigning positive predictions to samples with $\boldsymbol{x} > 0$ and negative predictions to $\boldsymbol{x} \leq 0$ minimizes the probability of error (Figure 3 **Right**). This optimal model is unique. Adding a fairness constraint that requires Mean EO $\leq 0.1$, the previously optimal classifier at 0 (with Mean EO $= 0.15$, **Right**) does not meet the fairness criteria. Searching over the threshold classifiers that minimize the probability of error while satisfying Mean EO constraint yields two equally optimal models (red and blue dots **Right**) with distinct decision regions (red and blue arrows **Left**). Even in this simple hypothesis class, the addition of fairness constraints yields multiple models with indistinguishable fairness and accuracy but with distinct decision regions. The arbitrary selection between these points can lead to arbitrary outputs to points near the boundary. ♦

## 4   Ensemble Algorithm for Arbitrariness Reduction

To tackle the potential arbitrariness cost of fairness intervention algorithms, we present a disparity-reduction mechanism through ensembling. We provide theoretical guarantees and numerical benchmarks to demonstrate that this method significantly reduces the predictive multiplicity of fair and accurate models.

In a nutshell, given competing models $h_1, \cdots, h_m$, we argue that the disparity in their score assignment can be reduced by considering a convex combination of them, defined as follows.

**Definition 7** (Ensemble Classifier). Given $m$ classifiers $\{h_1, \cdots, h_m : \mathbb{R}^d \to [0, 1]\}$ and a vector $\boldsymbol{\lambda} \in \boldsymbol{\Delta}_m$, we define the $\boldsymbol{\lambda}$-*ensembling* of the $h_j$ to be the convex combination $\boldsymbol{h}^{\mathrm{ens}, \boldsymbol{\lambda}} \triangleq \sum_{j \in [m]} \lambda_j h_j$.

## 4.1 Concentration of Ensembled Scores

We prove in the following result that *any* two different ensembling methods agree for fixed individuals with high probability. Recall that we fix a dataset $\mathcal{D}$ and a set of competing models $\mathcal{T}(\mathcal{D})$ coming from a stochastic training algorithm $\mathcal{T}$ (see Section 2). All proofs are provided in Appendix B.

**Theorem 4.1** (Concentration of Ensembles' Scores). *Let* $h_1, \ldots, h_m; \widetilde{h}_1, \ldots, \widetilde{h}_m \overset{iid}{\sim} \mathcal{T}(\mathcal{D})$ *be* $2m$ *models drawn from* $\mathcal{T}(\mathcal{D})$, *and* $\boldsymbol{h}^{\mathrm{ens}, \boldsymbol{\lambda}}, \widetilde{\boldsymbol{h}}^{\mathrm{ens}, \boldsymbol{\gamma}}$ *be the ensembled models (constructed with* $\{h_1, \ldots, h_m\}$ *and* $\{\widetilde{h}_1, \ldots, \widetilde{h}_m\}$ *respectively) for* $\boldsymbol{\lambda}, \boldsymbol{\gamma} \in \boldsymbol{\Delta}_m$ *(see Definition 7) satisfying* $\|\boldsymbol{\lambda}\|_2^2, \|\boldsymbol{\gamma}\|_2^2 \leq c/m$ *for an absolute constant c. For every* $\boldsymbol{x} \in \mathbb{R}^d$ *and* $\nu \geq 0$, *we have the exponentially-decaying (in m) bound*

$$\mathbb{P}\left(\left|\boldsymbol{h}^{\mathrm{ens}, \boldsymbol{\lambda}}(\boldsymbol{x}) - \widetilde{\boldsymbol{h}}^{\mathrm{ens}, \boldsymbol{\gamma}}(\boldsymbol{x})\right| \geq \nu\right) \leq 4e^{-\nu^2 m/(2c)}. \tag{7}$$

*In particular, for any validation set* $\mathcal{D}_{\mathrm{valid.}} \subset \mathbb{R}^d$ *of size* $|\mathcal{D}_{\mathrm{valid.}}| = n$, *we have the uniform bound*

$$\mathbb{P}\left(\left|\boldsymbol{h}^{\mathrm{ens}, \boldsymbol{\lambda}}(\boldsymbol{x}) - \widetilde{\boldsymbol{h}}^{\mathrm{ens}, \boldsymbol{\gamma}}(\boldsymbol{x})\right| < \nu \text{ for all } \boldsymbol{x} \in \mathcal{D}_{\mathrm{valid.}}\right) > 1 - 4ne^{-\nu^2 m/(2c)}. \tag{8}$$

## 4.2 Concentration of Predictions Under Ensembling

The above theorem implies that we can have a dataset of size that is exponential in the number of accessible competing models and still obtain similar scoring for *any* two ensembled models (uniformly across the dataset).

In practice, one cares more about the agreement of the final prediction of the classifiers. The following result extends Theorem 4.1 to the concentration of thresholded classifiers. For this, we need to define the notion of *confident classifiers*.

**Definition 8** (Confident Classifier). Fix a probability measure $P_{\boldsymbol{X}}$ over $\mathbb{R}^d$ and constants $\delta, \theta \in [0, 1]$. We say that a classifier $h : \mathbb{R}^d \to [0, 1]$ is $(P_{\boldsymbol{X}}, \delta, \theta)$-*confident* if $\mathbb{P}\left(|h(\boldsymbol{X}) - \frac{1}{2}| < \delta\right) < \theta$.

In other words, $h$ is a confident classifier if it is "more sure" of its predictions. We observe in experiments that models corrected by fairness interventions have scores concentrated around 0 and 1.

Using confident classifiers, we are able to extend Theorem 4.1 to thresholded ensembles, as follows.

**Theorem 4.2.** *Let* $\boldsymbol{h}^{\mathrm{ens}, \boldsymbol{\lambda}}, \widetilde{\boldsymbol{h}}^{\mathrm{ens}, \boldsymbol{\gamma}}$ *be as in Theorem 4.1, and assume that both ensembled classifiers are* $(P_{\boldsymbol{X}}, \delta, \theta)$-*confident in the sense of Definition 8. Let* $f(t) \triangleq \mathbb{1}\{t \geq 0.5\}$ *be the thresholding function. For any set* $\mathcal{D}_{\mathrm{valid.}} \subset \mathbb{R}^d$ *of size* $|\mathcal{D}_{\mathrm{valid.}}| = n$, *we may guarantee the probability of agreement in the predictions for all samples under the two ensembles to be at least*

$$\mathbb{P}\left(f(\boldsymbol{h}^{\mathrm{ens}, \boldsymbol{\lambda}}(\boldsymbol{x})) = f(\widetilde{\boldsymbol{h}}^{\mathrm{ens}, \boldsymbol{\gamma}}(\boldsymbol{x})) \text{ for every } \boldsymbol{x} \in \mathcal{D}_{\mathrm{valid.}}\right) \geq 1 - \left(4e^{-2\delta^2 m/c} + 2\theta\right)n. \tag{9}$$

We note that in the fairness-intervention setting, the set $\mathcal{D}_{\mathrm{valid.}}$ in the above theorem would be chosen as the subset of samples having the same group attribute. Thus, the size $n_0$ of $\mathcal{D}_{\mathrm{valid.}}$ would be significantly smaller than the total size of the dataset, and the parameter $\theta$ then can be required to be moderately small.

**Remark 2.** In Appendix C, we discuss how to optimize the ensembling parameters $\boldsymbol{\lambda}$. In the next section, we will stick to the uniform ensembling: $\boldsymbol{h}^{\mathrm{ens}, \boldsymbol{\lambda}} = \frac{1}{m} \sum_{j \in [m]} h_j$, i.e., $\boldsymbol{\lambda} = \frac{1}{m}\boldsymbol{1}$. This simple uniform ensemble suffices to illustrate the main goal of this paper: that arbitrariness can be a by-product of fairness intervention methods, and ensembling can mitigate this unwanted effect.

# 5 Experimental Results

We present empirical results to show that arbitrariness is masked by favorable group-fairness and accuracy metrics for multiple fairness intervention methods, baseline models, and datasets [7]. We also demonstrate the effectiveness of the ensemble in reducing the predictive multiplicity of fair models.

---

[7] Code can be found at https://github.com/Carol-Long/Fairness_and_Arbitrariness

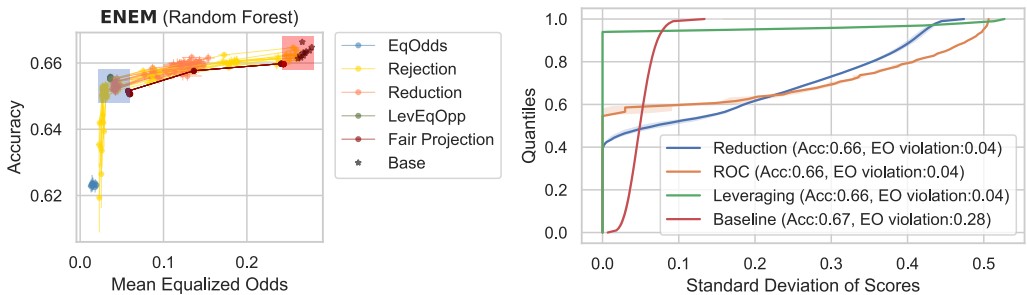

**Figure 4:** Quantile plot on high-fairness bin for various fairness interventions v.s. baseline on ENEM. **Left**: Fairness-Accuracy frontier. **Right**: Fair models produce larger score std. at top percentiles compared to the baseline model (horizontal axis computed via (6)). (REJECTION and LEVERAGING output thresholded scores directly.)

**Setup and Metrics.** We consider three baseline classifiers (BASE): random forest (RF), gradient boosting (GBM), and logistic regression (LR), implemented by Scikit-learn [36]. By varying the random seed, we obtain 10 baseline models with comparable performance. Then, we apply various state-of-the-art fairness methods (details in Appendix D) on the baseline models to get competing fair models.

On the test set, we compute mean accuracy, Mean EO (Definition 6), and predictive multiplicity levels on competing models before and after fairness interventions. We use ambiguity (Definition 2) and score standard deviations (Definition 3) as metrics for predictive multiplicity.

**Datasets.** We report predictive multiplicity and benchmark the ensemble method on three datasets – two datasets in the education domain: the high-school longitudinal study (HSLS) dataset [27, 28] and the ENEM dataset [16] (see Alghamdi et al. [2] Appendix B.1), and the UCI Adult dataset[33] which is based on the US census income data. The ENEM dataset contains Brazilian college entrance exam scores along with student demographic information and socio-economic questionnaire answers (e.g. if they own a computer). After pre-processing, the dataset contains 1.4 million samples with 139 features. Race is used as the group attribute $S$, and Humanities exam score is used as the label $Y$. Scores are quantized into two classes for binary classification. The race feature $S$ is binarized into White and Asian ($S = 1$) and others ($S = 0$). The experiments are run with a smaller version of the dataset with 50k samples. Complete experimental results can be found in Appendix E.

**Results that Reveal Arbitrariness.** We juxtapose the fairness-accuracy frontier and metrics for predictive ambiguity to reveal arbitrariness masked by favorable group-fairness and accuracy metrics in Figure 1 and 4. Starting with 10 baseline classifiers by varying the random seed used to initialize the training algorithm, we apply the fair interventions REDUCTION [1], REJECTION [29], LEVERAGING [12] to obtain point clouds of models with comparable fairness and accuracy metrics. In Figure 4, we take models that achieve very favorable accuracy and MEO metrics (in blue rectangle in **Left**) and plot the std. of scores to illustrate predictive multiplicity **Right**. Group fairness violations are greatly reduced (from 0.28 in baseline to 0.04 in fair models) at a small accuracy cost (from 67% in baseline to 66% in fair models). However, there is higher arbitrariness.

Compared to baseline (red curve), fair models corrected by REDUCTION and ROC produce lower score arbitrariness for the bottom 50% but much higher arbitrariness for the top 50% of samples; importantly, the induced arbitrariness becomes *highly nonuniform across different individuals* after applying the two fairness intervention. We observe that LEVERAGING produce models that agree on 90% of the samples, thereby not inducing concerns of arbitrariness.

Remarkably, arbitrariness does not vary significantly among models with different fairness levels. We consider two sets of models trained with high and low fairness constraints using REDUCTION in Figure 1.

**Results on the Effectiveness of Ensembling.** We pair our proofs in Section 4 with experiments that demonstrate the concentration of scores of ensembled models. In Figure 5 **Left**, taking the competing

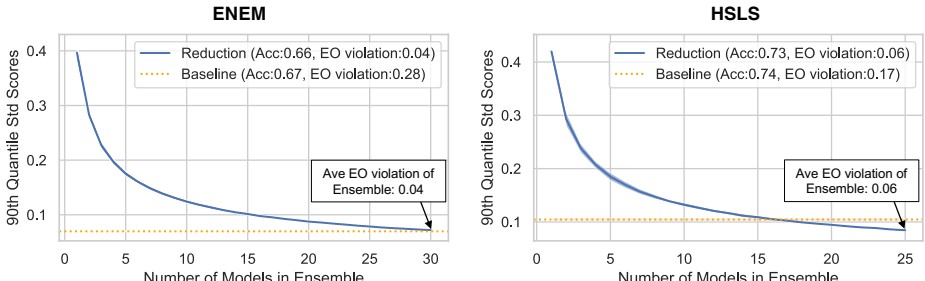

**Figure 5:** Standard deviation of ensembled models trained on ENEM and HSLS with baseline random forest classifiers. We fix the high-fairness bin and vary the number of models $m$ in each ensemble. As we increase the number of ensembles, score std. (on 10 ensembles) drops and meets the score std. of 10 baseline RFC when $m = 30$ on ENEM and $m = 17$ on HSLS. (Mean EO is computed using (6).

models in the high-fairness bins corrected with REDUCTION that achieve an Mean EO violation of 0.04 but very high score std. for half of the samples (blue rectangle in Figure 4), we ensemble the models with increasing number of models per ensemble ($m$) ranging from 1 to 30. For each $m$, we measure std. of scores in 10 such ensembles. The top percentile std. of the ensembled fair models drops to baseline with 30 models. Similar convergence occur on the HSLS dataset. Importantly, the ensembled models are still fair, the Mean EO violations of the ensembled models remain low.

## 6 Final Remarks

We demonstrate in this paper that arbitrariness is a facet of responsible machine learning that is orthogonal to existing fairness-accuracy analyses. Specifically, fairness-vs-accuracy frontiers are insufficient for detecting arbitrariness in the predictions of group-fair models: two models can have the same fairness-accuracy curve while at the same time giving widely different predictions for subsets of individuals. We demonstrate this undesirable phenomenon both theoretically and experimentally on state-of-the-art fairness intervention methods. Furthermore, towards mitigating this arbitrariness issue, we propose an ensemble algorithm, where a convex combination of several competing models is used for decision-making instead of any of the constituent models. We prove that the scores of the ensemble classifier concentrate, and that the ensuing predictions can be made to concentrate under mild assumptions. Importantly, we exhibit via real-world experiments that our proposed ensemble algorithm can reduce arbitrariness while maintaining fairness and accuracy.

**Limitations.** The proposed framework for estimating the predictive multiplicity of fairness interventions requires re-training multiple times, limiting its applicability to large models. We consider model variation due to randomness used during training. In practice, competing models may exist due to inherent uncertainty (i.e., a non-zero confidence interval) when evaluating model performance on a finite test set. In this regard, models with comparable average performance can be produced by searching over this Rashomon set even if training is deterministic (e.g., equivalent to solving a convex optimization problem).

**Future directions.** An interesting future direction is to explore the multiplicity cost of fairness interventions in such deterministic settings. Furthermore, our ensembling strategy may not guarantee that the ensemble classifier ensures fairness constraints due to the non-convex nature of such constraints. Though we empirically observe that fairness constraints are indeed satisfied by the ensemble model, proving such guarantees theoretically would be valuable.

**Societal impacts.** While fairness intervention algorithms can effectively reduce the disparate impact among population groups, they can induce predictive multiplicity in individual samples. The increase in predictive multiplicity does not impact all individuals equally. If predictive multiplicity caused by fairness interventions is not accounted for, some individuals will bear the brunt of arbitrary decision-making—their predictions could be arbitrary upon re-training the classifier using different random initializations, leading to another level of disparate treatment to certain population groups.

**Acknowledgements.** The authors would like to thank Jamelle Watson-Daniels, Arpita Biswas and Bogdan Kulynych for hepful discussions on initial ideas. This material is based upon work supported by the National Science Foundation under grants CAREER 1845852, CIF 1900750, CIF 2312667 and FAI 2040880, and by Meta Ph.D. fellowship.

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

# Appendix

The appendix is divided into the following five parts. Appendix A: proof of the orthogonality of OAE and SP from ambiguity (Proposition 3.1); Appendix B: proofs from Section 4; Appendix C: discussion on optimizing ensemble parameters; Appendix D: additional discussions on group fairness and fairness interventions; and Appendix E: additional experiments and details on the experimental setup.

## A  Proof of Proposition 3.1: Orthogonality of OAE and SP from Ambiguity

We divide the proof into two cases according to the group-fairness metric considered (OAE or SP).

### A.1  Proof for the OAE Metric

Let $\mathcal{D} \subset \mathbb{R}^d \times [K] \times \{0,1\}$ be a dataset satisfying the two conditions listed in Remark 1, namely,

1. with $n_k$ denoting the number of samples in $\mathcal{D}$ belonging to group $k \in [K]$, the greatest common divisor of the $n_k$ is at least $(m-1)/(m\epsilon - 1)$.
2. if $(\boldsymbol{x}, s, y), (\boldsymbol{x}, s', y') \in \mathcal{D}$ share the same feature vector $\boldsymbol{x}$, then $y = y'$ too.

Consider the partition along group membership $\mathcal{D} = \bigcup_{k \in [K]} \mathcal{D}_k$, so we may write $\mathcal{D}_k = \{(\boldsymbol{x}_{k,i}, k, y_{k,i})\}_{i \in [n_k]}$ for each $k \in [K]$, where $n_k \triangleq |\mathcal{D}_k|$. By assumption on the $n_k$, we have

$$g \triangleq \gcd(n_1, \cdots, n_K) \geq \frac{m-1}{m\epsilon - 1}. \tag{10}$$

Now, let $\theta \triangleq \lfloor \epsilon g \rfloor / g$, and note that we obtain the integers $\mu_k \triangleq \theta \cdot n_k$ for $k \in [K]$, because $g$ divides each $n_k$ by definition of the gcd. Let $\lambda \triangleq \lceil \frac{1-\epsilon}{\theta} \rceil$. Using $\lfloor t \rfloor > t - 1$, we note that

$$\frac{1 - \epsilon}{\theta} + 1 < \frac{1 - \epsilon}{\epsilon - \frac{1}{g}} + 1 \leq m, \tag{11}$$

where the last inequality follows by assumption on $g$ being large enough (10). Thus, taking the ceiling of both sides above, we obtain $\lambda + 1 \leq m$. Hence, it suffices to prove that there are $\lambda + 1$ classifiers $\mathcal{R} = \{h_j\}_{j \in [\lambda + 1]}$ satisfying the claim in the proposition (i.e., with $\lambda + 1$ replacing $m$).

It is straightforward to check that $\lambda \mu_k \leq n_k$ for each $k$. We divide each $\mathcal{D}_k$ into $\lambda$ sets of size $\mu_k$ each, and collect the remainder into a separate subset. Thus, for each $k \in [K]$, consider any partition $\mathcal{D}_k = \bigcup_{j \in [\lambda + 1]} \mathcal{D}_k^j$ where $|\mathcal{D}_k^j| = \mu_k$ for each $j \in [\lambda]$ and $|\mathcal{D}_k^{\lambda+1}| = n_k - \lambda \mu_k$.

Next, we define the classifiers based on the partitions of the $\mathcal{D}_k$. For each $j \in [\lambda + 1]$, define the classifier $h_j$ over $\mathcal{D}$ as follows. Fix $(\boldsymbol{x}, k, y) \in \mathcal{D}$. We set the value $h_j(\boldsymbol{x})$ to be

$$h_j(\boldsymbol{x}) \triangleq \begin{cases} 1 - y & \text{if } (\boldsymbol{x}, k, y) \in \mathcal{D}_k^j, \\ y & \text{otherwise.} \end{cases} \tag{12}$$

Note that this makes $h_j$ well-defined by the second property assumed on $\mathcal{D}$ at the beginning of this proof. We show that the set $\mathcal{R} = \{h_j\}_{j \in [\lambda + 1]}$ satisfies the desired properties in the proposition.

**Accuracy.**  First, we show that each $h_j$ incurs a 0-1 loss less than $\epsilon$. Indeed, we have that, for each $j \in [\lambda]$,

$$\ell(h_j; \mathcal{D}) = \frac{1}{n} \sum_{k \in [K]} |\mathcal{D}_k^j| = \frac{1}{n} \sum_{k \in [K]} \mu_k = \theta, \tag{13}$$

whereas the error for the last classifier is

$$\ell(h_{\lambda+1}; \mathcal{D}) = \frac{1}{n} \sum_{k \in [K]} |\mathcal{D}_k^{\lambda+1}| = \frac{1}{n} \sum_{k \in [K]} n_k - \lambda \mu_k = 1 - \lambda \theta. \tag{14}$$

Thus, it suffices to check that $\theta, 1 - \lambda\theta \leq \epsilon$. Recall that we set $\theta = \lfloor \epsilon g \rfloor / g$, hence $\theta \leq \epsilon$ is immediate. Further, as we take $\lambda = \lceil \frac{1-\epsilon}{\theta} \rceil \geq \frac{1-\epsilon}{\theta}$, the inequality $1 - \lambda\theta \leq \epsilon$ follows immediately too. Hence, we have that $\ell(h_j; \mathcal{D}) \leq \epsilon$ for all $j \in [\lambda + 1]$.

**Fairness.** Next, we check that each $h_j$ satisfies Overall Accuracy Equality (OAE) perfectly. Let $\widehat{Y}_j$ be the prediction of the classifier $h_j$, so $P_{\widehat{Y}_j|\boldsymbol{X}=\boldsymbol{x}}(1) = h_j(\boldsymbol{x})$. Then, for $j \in [\lambda]$ the predictions of the $j$-th classifier satisfy

$$\Pr\left(\widehat{Y}_j \neq Y \mid S = k\right) = \frac{|\mathcal{D}_k^j|}{n_k} = \frac{\mu_k}{n_k} = \theta \qquad \text{for every group } k \in [K], \tag{15}$$

and similarly the last classifier satisfies

$$\Pr\left(\widehat{Y}_{\lambda+1} \neq Y \mid S = k\right) = \frac{|\mathcal{D}_k^{\lambda+1}|}{n_k} = \frac{n_k - \lambda\mu_k}{n_k} = 1 - \lambda\theta \qquad \text{for every group } k \in [K]. \tag{16}$$

Hence, the $h_j$, for $j \in [\lambda+1]$, all satisfy OAE perfectly.

**Ambiguity.** Finally, we show that the set $\mathcal{R} = \{h_j\}_{j \in [\lambda+1]}$ exhibits full ambiguity. Note that we have a partition $\mathcal{D} = \bigcup_{(k,j) \in [K] \times [\lambda+1]} \mathcal{D}_k^j$. Fix a sample $(\boldsymbol{x}, s, y) \in \mathcal{D}$ and consider the index $(k, j) \in [K] \times [\lambda+1]$ of the unique part including it in the partition, i.e., $(\boldsymbol{x}, s, y) \in \mathcal{D}_k^j$. By construction of $\mathcal{R}$, we have that $h_j(\boldsymbol{x}) = 1 - y$ (so $\widehat{Y}_j = 1 - y$) but $h_{j'}(\boldsymbol{x}) = y$ (so $\widehat{Y}_{j'} = y$) for any $j' \neq j$. In other words, for *every* fixed sample in $\mathcal{D}$, there is a pair of classifiers in $\mathcal{R}$ assigning it conflicting predictions. Thus, the ambiguity is $\alpha(\mathcal{D}, \mathcal{R}) = 100\%$.

## A.2 Proof for the SP Metric

Consider now the SP group-fairness constraint instead. In this case, we define we construct the following alternative dataset $\mathcal{D} \subset \mathbb{R}^d \times [K] \times \{0, 1\}$. Let $t \triangleq \lceil 1/\epsilon \rceil \geq 2$. For each $(k, y) \in [K] \times \{0, 1\}$, fix any dataset $\mathcal{D}_{k,y} = \{(\boldsymbol{x}_{k,y,i}, k, y)\}_{i \in [t]} \subset \mathbb{R}^d \times \{k\} \times \{y\}$, let $\mathcal{D} \triangleq \bigcup_{(k,y) \in [K] \times \{0,1\}} \mathcal{D}_{k,y} \subset \mathbb{R}^d \times [K] \times \{0, 1\}$, and assume that if $\boldsymbol{x}_{k,y,i} = \boldsymbol{x}_{k',y',i'}$ then $y = y'$. Denote $n \triangleq |\mathcal{D}| = 2Kt$.

For each $(k, y) \in [K] \times \{0, 1\}$, let $\sigma_{k,y}$ be a permutation on $[t]$, i.e., $\{\sigma_{k,y}(j)\}_{j \in [t]} = [t]$. Define the set of classifiers $\mathcal{R} = \{h_j\}_{j \in [t]}$ on $\mathcal{D}$ as follows. For each $j \in [t]$ and each $(k, y, i) \in [K] \times \{0, 1\} \times [t]$, we set

$$h_j(\boldsymbol{x}_{k,y,i}) \triangleq \begin{cases} 1 - y & \text{if } i = \sigma_{k,y}(j), \\ y & \text{otherwise.} \end{cases} \tag{17}$$

We show that the set $\mathcal{R} = \{h_j\}_{j \in [t]}$ satisfies the desired properties in the proposition.

**Accuracy.** Consider first the 0-1 loss incurred by the $h_j$. We have that, for each $j \in [t]$,

$$\ell(h_j; \mathcal{D}) = \frac{1}{n} \sum_{(k,y,i) \in [K] \times \{0,1\} \times [t]} \mathbb{1}\{i = \sigma_{k,y}(j)\} = \frac{1}{n} \sum_{(k,y) \in [K] \times \{0,1\}} 1 = \frac{2K}{2Kt} = \frac{1}{t}. \tag{18}$$

Therefore, $\ell(h_j; \mathcal{D}) = 1/t = 1/\lceil 1/\epsilon \rceil \leq \epsilon$. In other words, $\mathcal{R}$ is a realization of the $\epsilon$-level empirical Rashomon set of size $t$.

**Fairness.** Next, we check that each $h_j$ satisfies SP perfectly. In other words, we check that $h_j$ assigns to the class $y = 1$ the same percentage of samples across the groups $k \in [K]$. Indeed, this is true as we are switching the class memberships of exactly one sample from each of the $\mathcal{D}_{k,y}$. In particular, $h_j$ assigns the class membership 1 to exactly $1 + (|\mathcal{D}_{k,1}| - 1) = |\mathcal{D}_{k,1}| = t$ samples out of the total $|\mathcal{D}_{k,0} \cup \mathcal{D}_{k,1}| = 2t$ samples belonging group $k$. As the ratio $t/(2t) = 1/2$ is independent of the group $k$, we obtain the desired result that $h_j$ satisfies SP.

**Ambiguity.** Finally, we check that the set $\mathcal{R} = \{h_j\}_{j \in [t]}$ suffers from 100% ambiguity. Fix a sample $(\boldsymbol{x}_{k,y,i}, k, y) \in \mathcal{D}$, and we will show that there are two classifiers in $\mathcal{R}$ assigning conflicting predictions to it. Indeed, let $j = \sigma_{k,y}^{-1}(i)$ and let $j' \in [t] \setminus \{j\}$ be any other index. Then, $h_j(\boldsymbol{x}_{k,y,i}) = 1 - y$, whereas $h_{j'}(\boldsymbol{x}_{k,y,i}) = y$. Therefore, the sample $(\boldsymbol{x}_{k,y,i}, k, y)$ contributes to the overall ambiguity $\alpha(\mathcal{D}, \mathcal{R})$. As this is true for all samples in $\mathcal{D}$, we conclude that $\alpha(\mathcal{D}, \mathcal{R}) = 100\%$, as desired. This completes the proof of the proposition.

# B  Proofs of Section 4

## B.1  Proof of Theorem 4.1

We assume that $\|\boldsymbol{\lambda}\|_2^2, \|\boldsymbol{\gamma}\|_2^2 \leq c/m$ for an absolute constant $c$, e.g., we have $c = 1$ for the uniform ensembling $\boldsymbol{\lambda} = (1/m, \cdots, 1/m)$ as then $\|\boldsymbol{\lambda}\|_2^2 = 1/m$. Fix $\boldsymbol{x}$, and denote the mean of the classifiers $\mu_{\boldsymbol{x}} = \mathbb{E}_{h \sim \mathcal{T}(\mathcal{D})}[h(\boldsymbol{x})]$. The mapping $(h_1, \cdots, h_m) \mapsto \boldsymbol{h}^{\mathrm{ens},\boldsymbol{\lambda}}$ satisfies the bounded-difference condition in the McDiarmid inequality. Indeed, changing $h_i$ can change $\boldsymbol{h}^{\mathrm{ens},\boldsymbol{\lambda}}$ by at most $\lambda_i$. Furthermore, $\boldsymbol{h}^{\mathrm{ens},\boldsymbol{\lambda}}(\boldsymbol{x})$ has the mean

$$\mathbb{E}\left[\boldsymbol{h}^{\mathrm{ens},\boldsymbol{\lambda}}(\boldsymbol{x})\right] = \sum_{i \in [m]} \lambda_i \mathbb{E}[h_i(\boldsymbol{x})] = \mu_{\boldsymbol{x}} \sum_{i \in [m]} \lambda_i = \mu_{\boldsymbol{x}}. \tag{19}$$

Hence, by Mcdiarmid's inequality, we have the bound

$$\mathbb{P}\left(\left|\boldsymbol{h}^{\mathrm{ens},\boldsymbol{\lambda}}(\boldsymbol{x}) - \mu_{\boldsymbol{x}}\right| \geq \nu\right) \leq 2\exp\left(\frac{-2\nu^2}{\sum_{i \in [m]} \lambda_i^2}\right) \leq 2\exp\left(\frac{-2\nu^2 m}{c}\right). \tag{20}$$

The same inequality holds for $\boldsymbol{\gamma}$ in place of $\boldsymbol{\lambda}$:

$$\mathbb{P}\left(\left|\widetilde{\boldsymbol{h}}^{\mathrm{ens},\boldsymbol{\gamma}}(\boldsymbol{x}) - \mu_{\boldsymbol{x}}\right| \geq \nu\right) \leq 2\exp\left(\frac{-2\nu^2 m}{c}\right). \tag{21}$$

Therefore, we obtain the bound

$$1 - \mathbb{P}\left(\left|\boldsymbol{h}^{\mathrm{ens},\boldsymbol{\lambda}}(\boldsymbol{x}) - \widetilde{\boldsymbol{h}}^{\mathrm{ens},\boldsymbol{\gamma}}(\boldsymbol{x})\right| \geq \nu\right) = \mathbb{P}\left(\left|\boldsymbol{h}^{\mathrm{ens},\boldsymbol{\lambda}}(\boldsymbol{x}) - \mu_{\boldsymbol{x}} + \mu_{\boldsymbol{x}} - \widetilde{\boldsymbol{h}}^{\mathrm{ens},\boldsymbol{\gamma}}(\boldsymbol{x})\right| < \nu\right) \tag{22}$$

$$\geq \mathbb{P}\left(\left|\boldsymbol{h}^{\mathrm{ens},\boldsymbol{\lambda}}(\boldsymbol{x}) - \mu_{\boldsymbol{x}}\right| + \left|\widetilde{\boldsymbol{h}}^{\mathrm{ens},\boldsymbol{\gamma}}(\boldsymbol{x}) - \mu_{\boldsymbol{x}}\right| < \nu\right) \tag{23}$$

$$\geq \mathbb{P}\left(\left|\boldsymbol{h}^{\mathrm{ens},\boldsymbol{\lambda}}(\boldsymbol{x}) - \mu_{\boldsymbol{x}}\right| < \frac{\nu}{2} \cap \left|\widetilde{\boldsymbol{h}}^{\mathrm{ens},\boldsymbol{\gamma}}(\boldsymbol{x}) - \mu_{\boldsymbol{x}}\right| < \frac{\nu}{2}\right) \tag{24}$$

$$= 1 - \mathbb{P}\left(\left|\boldsymbol{h}^{\mathrm{ens},\boldsymbol{\lambda}}(\boldsymbol{x}) - \mu_{\boldsymbol{x}}\right| \geq \frac{\nu}{2} \cup \left|\widetilde{\boldsymbol{h}}^{\mathrm{ens},\boldsymbol{\gamma}}(\boldsymbol{x}) - \mu_{\boldsymbol{x}}\right| \geq \frac{\nu}{2}\right) \tag{25}$$

$$\geq 1 - 4\exp\left(\frac{-\nu^2 m}{2c}\right), \tag{26}$$

where the first inequality comes from triangle inequality, the following from probability of subset of events ($\mathbb{P}(A) \geq \mathbb{P}(B)$ if $A \supseteq B$), the equality from taking complement, and the last line from applying McDiarmid's inequality along with the union bound.

Finally, applying the union bound on $\mathcal{D}_{\mathrm{valid.}}$ with $|\mathcal{D}_{\mathrm{valid.}}| = n$, we obtain the bound

$$\mathbb{P}\left(\bigcap_{\boldsymbol{x} \in \mathcal{D}_{\mathrm{valid.}}} \left|\boldsymbol{h}^{\mathrm{ens},\boldsymbol{\lambda}}(\boldsymbol{x}) - \widetilde{\boldsymbol{h}}^{\mathrm{ens},\boldsymbol{\gamma}}(\boldsymbol{x})\right| < \nu\right) = 1 - \mathbb{P}\left(\bigcup_{\boldsymbol{x} \in \mathcal{D}_{\mathrm{valid.}}} \left|\boldsymbol{h}^{\mathrm{ens},\boldsymbol{\lambda}}(\boldsymbol{x}) - \widetilde{\boldsymbol{h}}^{\mathrm{ens},\boldsymbol{\gamma}}(\boldsymbol{x})\right| \geq \nu\right) \tag{27}$$

$$\geq 1 - 4n\exp\left(\frac{-\nu^2 m}{2c}\right), \tag{28}$$

and the proof is complete.

## B.2  Proof of Theorem 4.2

The main idea is as follows: first observe that for the ensembled labels to disagree on a sample (given that the scores are bounded away from $\frac{1}{2}$ with high probability), the two models need to produce scores in the range $[0, \frac{1}{2} - \delta] \cup [\frac{1}{2} + \delta, 1]$. This means that the scores need to deviate at least $2\delta$ which has an exponentially low probability given Theorem 4.1.

We will show that

$$\mathbb{P}\left(\bigcup_{\boldsymbol{x} \in \mathcal{D}_0} f(\boldsymbol{h}^{\mathrm{ens},\boldsymbol{\lambda}}(\boldsymbol{x})) \neq f(\widetilde{\boldsymbol{h}}^{\mathrm{ens},\boldsymbol{\gamma}}(\boldsymbol{x}))\right) \leq \left(4e^{-2\delta^2 m/c} + 2\theta\right) n_0. \tag{29}$$

Indeed, for each fixed $\boldsymbol{x} \in \mathcal{D}_0$, we may reduce the failure probability to the case of separation of scores:

$$\mathbb{P}\left(f(\boldsymbol{h}^{\mathrm{ens},\boldsymbol{\lambda}}(\boldsymbol{x})) \neq f(\widetilde{\boldsymbol{h}}^{\mathrm{ens},\boldsymbol{\gamma}}(\boldsymbol{x}))\right) \leq \mathbb{P}\Bigg( \left(\boldsymbol{h}^{\mathrm{ens},\boldsymbol{\lambda}}(\boldsymbol{x}) \in \left[0, \frac{1}{2}-\delta\right] \cap \widetilde{\boldsymbol{h}}^{\mathrm{ens},\boldsymbol{\gamma}}(\boldsymbol{x}) \in \left[\frac{1}{2}+\delta, 1\right]\right) \tag{30}$$

$$\cup \left(\widetilde{\boldsymbol{h}}^{\mathrm{ens},\boldsymbol{\gamma}}(\boldsymbol{x}) \in \left[0, \frac{1}{2}-\delta\right] \cap \boldsymbol{h}^{\mathrm{ens},\boldsymbol{\lambda}}(\boldsymbol{x}) \in \left[\frac{1}{2}+\delta, 1\right]\right) \tag{31}$$

$$\cup \boldsymbol{h}^{\mathrm{ens},\boldsymbol{\lambda}}(\boldsymbol{x}) \in \left[\frac{1}{2}-\delta, \frac{1}{2}+\delta\right] \tag{32}$$

$$\cup \widetilde{\boldsymbol{h}}^{\mathrm{ens},\boldsymbol{\gamma}}(\boldsymbol{x}) \in \left[\frac{1}{2}-\delta, \frac{1}{2}+\delta\right]\Bigg) \tag{33}$$

$$\leq \mathbb{P}\left(\left|\boldsymbol{h}^{\mathrm{ens},\boldsymbol{\lambda}}(\boldsymbol{x}) - \widetilde{\boldsymbol{h}}^{\mathrm{ens},\boldsymbol{\gamma}}(\boldsymbol{x})\right| \geq 2\delta\right) + 2\theta \tag{34}$$

$$\leq 4\exp\left(\frac{-2\delta^2 m}{c}\right) + 2\theta. \tag{35}$$

Finally, applying the union bound, we obtain that

$$\mathbb{P}\left(\bigcap_{\boldsymbol{x}\in\mathcal{D}_0} f(\boldsymbol{h}^{\mathrm{ens},\boldsymbol{\lambda}}(\boldsymbol{x})) = f(\widetilde{\boldsymbol{h}}^{\mathrm{ens},\boldsymbol{\gamma}}(\boldsymbol{x}))\right) = 1 - \mathbb{P}\left(\bigcup_{\boldsymbol{x}\in\mathcal{D}_0} f(\boldsymbol{h}^{\mathrm{ens},\boldsymbol{\lambda}}(\boldsymbol{x})) \neq f(\widetilde{\boldsymbol{h}}^{\mathrm{ens},\boldsymbol{\gamma}}(\boldsymbol{x}))\right) \tag{36}$$

$$\geq 1 - \left(4e^{-2\delta^2 m/c} + 2\theta\right)n_0, \tag{37}$$

and the proof is complete.

## C  Discussion on optimizing ensemble parameters

We have taken the weights $\boldsymbol{\lambda} \in \boldsymbol{\Delta}_m$ which determines the ensembled model $\boldsymbol{h}^{\mathrm{ens},\boldsymbol{\lambda}}$ to be fixed. We explain here how $\boldsymbol{\lambda}$ can be optimized according to a given cost. Specifically, given a loss function $\ell : [0,1] \times \{0,1\} \to \mathbb{R}_+$, we can search for the optimal $\boldsymbol{\lambda} \in \boldsymbol{\Delta}_m$ that minimizes the total cost

$$L_{\mathrm{ens}}(\boldsymbol{\lambda}) \triangleq \mathbb{E}\left[\ell(\boldsymbol{h}^{\mathrm{ens},\boldsymbol{\lambda}}(X), Y)\right]. \tag{38}$$

For the above optimization problem, we think of the constituent models $h_1, \cdots, h_m$ as being fixed and the randomness is from that of $(X, Y)$.

However, in practice, we have access to only samples $(\boldsymbol{x}_i, y_i) \sim P_{\boldsymbol{X},Y}$. Thus, we consider minimizing the regularized sample mean (for fixed $\beta > 0$)

$$\widehat{L}_{\mathrm{ens}}(\boldsymbol{\lambda}) \triangleq \frac{1}{n}\sum_{i\in[n]} \ell\left(\boldsymbol{h}^{\mathrm{ens},\boldsymbol{\lambda}}(\boldsymbol{x}_i), y_i\right) + \frac{\beta}{\sqrt{n}}\|\boldsymbol{\lambda}\|_2^2. \tag{39}$$

The 2-norm regularization is added to facilitate proving convergence. This convergence result can be obtained via known results from statistical learning theory, e.g., using Theorem 13.2 in [22]. Specifically, consider the following two restrictions:

- Consider only $\boldsymbol{\lambda} \in \boldsymbol{\Delta}_m$ satisfying $\|\boldsymbol{\lambda}\|_2 \leq \alpha$ for a prescribed $\alpha$. Note that we may take $\alpha = 1$ to encapsulate the whole probability simplex. However, we may choose $\boldsymbol{\lambda}$ to be a slight modification of the uniform ensembling, in which case we would have $\alpha$ of order $1/\sqrt{m}$.

- Assume that the function $\boldsymbol{\lambda} \mapsto \ell(\boldsymbol{h}^{\mathrm{ens},\boldsymbol{\lambda}}(\boldsymbol{x}), y)$ is convex and $A$-Lipschitz for each fixed $(\boldsymbol{x}, y)$.

In this case, choosing $\beta = A/\alpha$ and denoting the optimizers

$$\boldsymbol{\lambda}^{(n)} \triangleq \underset{\|\boldsymbol{\lambda}\|_2 \leq \alpha}{\mathrm{argmin}}\ \widehat{L}_{\mathrm{ens}}(\boldsymbol{\lambda}), \tag{40}$$

we can bound the utility of these minimizers by

$$\mathbb{P}\left(L_{\text{ens}}(\boldsymbol{\lambda}^{(n)}) \leq \inf_{\|\boldsymbol{\lambda}\|_2 \leq \alpha} L_{\text{ens}}(\boldsymbol{\lambda}) + \frac{\beta\alpha^2}{\sqrt{n}} \cdot \left(1 + \frac{1}{\delta} + \frac{8}{\delta\sqrt{n}}\right)\right) \geq 1 - \delta \qquad (41)$$

for any $\delta \in (0, 1)$.

## D   Additional discussion on group fairness and fairness interventions

In addition to OAE (Definition 5), two other important fairness criteria are Statistical Parity [18] and Equalized Odds [23].

**Definition 9** (SP). $\Pr(\widehat{Y} = 1|S = s) = \Pr(\widehat{Y} = 1|S = s')$ for all groups $s, s' \in [K]$ .

**Definition 10** (EO). $\Pr(\widehat{Y} = 1|S = s, Y = b) = \Pr(\widehat{Y} = 1|S = s', Y = y)$ for all groups $s, s' \in [K]$, and binary labels $y \in \{0, 1\}$.

Essentially, SP requires the predicted label $\widehat{Y} \triangleq \arg\max h(\boldsymbol{X})$ to be independent of the group membership $S$ [19]. In comparison, EO conditions on both group and the true label [23]. EO improves upon SP in the sense that it does not rule out the perfect classifiers whenever the true label $Y$ is correlated with the group membership $S$ [1]. In practice, we quantify EO violation by measuring Mean EO as in Equation 6 (for two groups) and, more generally, in Equation 42 below (beyond two groups). Similarly, we can measure SP violation as in Equation 43.

$$\text{MEAN EO} \triangleq \max_{s,s' \in [K]} \frac{1}{2}\left(|\text{TPR}_{S=s} - \text{TPR}_{S=s'}| + |\text{FPR}_{S=s} - \text{FPR}_{S=s'}|\right). \qquad (42)$$

$$\text{SP VIOLATION} \triangleq \max_{s,s' \in [K]} \frac{1}{2}\left(|\Pr(\widehat{Y} = 1|S = s) - \Pr(\widehat{Y} = 1|S = s')|\right). \qquad (43)$$

Next, we offer a brief discussion of various intervention mechanisms used in this paper. The fairness interventions can be categorized into two categories: in-processing and post-processing. In-processing mechanisms incorporate fairness constraints during training. It usually add the fairness constraint to the loss function and outputs a fair classifier. Post-processing mechanisms treat the model as a black box and update its predictions to achieve the desirable fairness constraints [10].

REDUCTION [1], short for exponentiated gradient reduction, is an in-processing technique that reduces fair classification to a sequence of cost-sensitive classification problems, and yields a randomized classifier with the lowest empirical error subject to the desired constraints. This technique achieves fairness with a minimal decrease in accuracy, but it is computationally expensive since it requires re-training multiple models.

REJECT OPTION CLASSIFIER [29] is a postprocessing technique that achieves fairness constraints by modifying outcomes of samples in a confidence band of the decision boundary with the highest uncertainty. It gives favorable outcomes to unprivileged groups and unfavorable outcomes to privileged groups. It outputs a thresholded prediction rather than a probability over the binary labels.

EQODDS [23] is a post-processing technique that formulates empirical risk minimization with fairness constraint as a linear program and modifies predictions according to the derived probabilities to achieve equalized odds.

FAIR PROJECTION [2] is a post-processing technique that can accommodate fairness constraints in a setting with multiple labels and multiple groups. The fair model is obtained from 'projecting' a pre-trained (and potentially unfair) classifier onto the set of models that satisfy target group-fairness requirements.

## E   Additional experiments and details on the experimental setup

Our proposed methodology can be summarized in the pipeline in Figure 6.

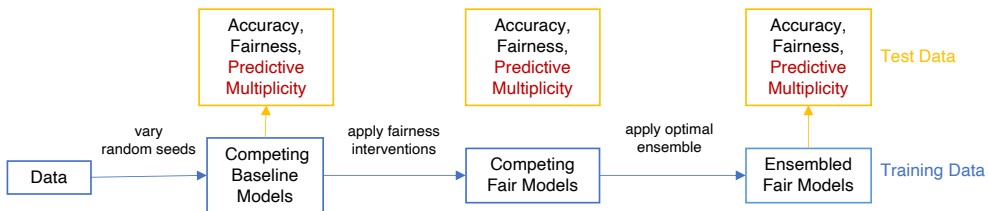

**Figure 6:** Flow chart of experimental procedure.

## E.1 Data

The HSLS dataset [27, 28] is an education dataset collected from 23,000+ students across high schools in the USA. Features of the dataset contain extensive information on students' demographic information, their parents' income and education level, schools' information, and students' academic performances across years. We apply the pre-processing techniques adopted by Alghamdi et al. [2], with the number of samples reduced to 14,509. For the binary classification task with fairness constraints, the group attribute chosen is RACE $\in$ {WHITE, NON-WHITE} and the prediction label is students 9th-Grade GRADEBIN $\in$ {0, 1}, binarized according to whether a student's grade is higher or lower than the median.

The ENEM dataset [16] is a Brazilian high school national exam dataset introduced by Alghamdi et al. [2]. It has 138 features containing students' demographic information, socio-economic questionnaire answers (e.g., parents' education level and if they own a computer), and students' exam scores. Adopting the preprocessing technique in Alghamdi et al. [2], we sample 50K samples without replacement from the processed ENEM Year 2020 data. Identical to HSLS, the group attribute chosen is RACE $\in$ {WHITE, NON-WHITE} and the prediction label is students Grade binarized into GRADEBIN $\in$ {0, 1} according to whether a student's grade is higher or lower than the median.

For the widely known Adult dataset [33], also known as "Census Income" dataset, we choose the group attribute as SEX $\in$ {MALE, FEMALE} and predicted label to be INCOME $\in$ {0, 1}, where income bin denotes whether a person's income is higher or lower than 50K/yr.

## E.2 Competing Baseline Models

We use the Scikit-learn implementation of logistic regression, gradient boosting, and random forest as baseline models. For logistic regression and gradient boosting, the default hyperparameter is used; for random forest, we set the number of trees and minimum number of samples per leaf to 10 to prevent over-fitting. To get 10 competing models for each hypothesis class, we use 10 random seeds (specifically 33–42).

In practice, the competing models, i.e., $h \in \widehat{\mathcal{R}}_m$ can be obtained using different methodologies, such as sampling and adversarial weight perturbation [26, 40]. We suggest one method for sampling. First, split the data into training, validation, and test dataset. We train a set of models by changing the randomized procedures in the training process, e.g., using different initializations, different cuts for cross-validation, data shuffling, etc. In this paper, we change the random seed feed into the baseline models to obtain competing models. We use the validation set to measure $\epsilon$ corresponding to this empirical Rashomon Set.

## E.3 Competing Fair Models

For EQODDS, REJECTION, and REDUCTION, we use the functions EQODDSPOSTPROCESSING, REJECTOPTIONCLASSIFICATION, and EXPONENTIATEDGRADIENTREDUCTION from AIF360 toolkits [4]. For LEVERAGING and FAIR PROJECTION, we use the codes provided in the corresponding Github repositories of Chzhen et al. [12] and Alghamdi et al. [2].

**Remark 3.** We observe in practice that fairness classifiers are more confident and have scores that are thresholded-like (Figure 7 **Left**). From the similarity in the shape of the thresholded baseline curve and the fair models' curves (Figure 7 **Right**), thresholding-like behavior of some interventions may explain some—but certainly not all (see Figure 1 **Right**)—increase in score std dev and the ensuing arbitrariness. Recall from the violin plot in Figure 1 that the largest group (blue area) are

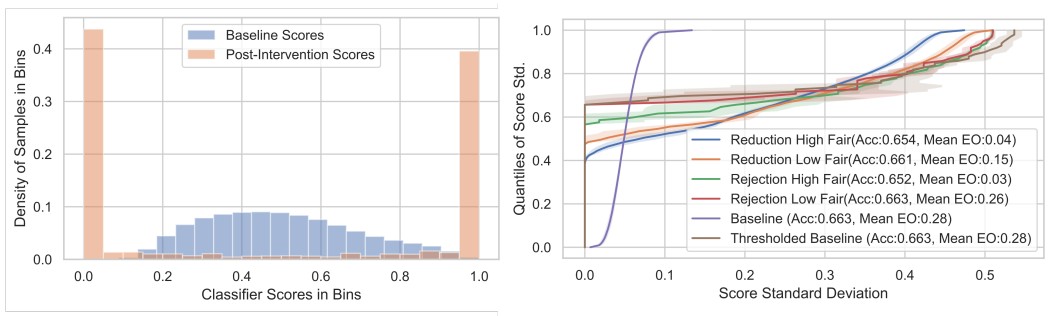

**Figure 7: Left**: We plot the score distribution of the REDUCTION approach as an example. The scores concentrate around 0 and 1, while scores of the baseline classifier are normally distributed. **Right**: Given this thresholded-like scores of fair classifiers, we include thresholded baseline in the quantile plot of score std.. The thresholded Baseline curve largely overlaps with Rejection curve with MEO 0.26 since they have the same MEO level.

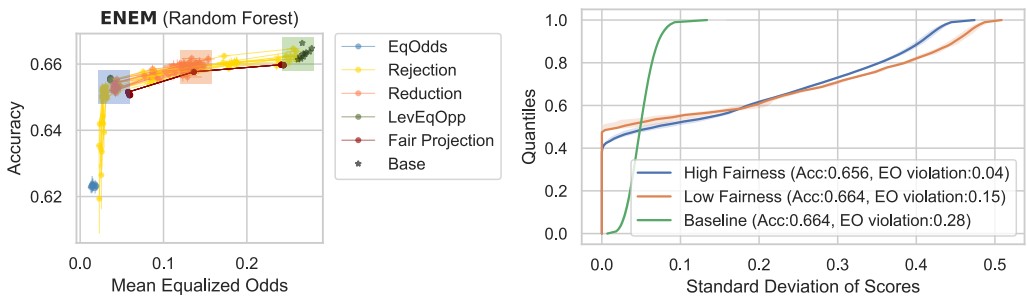

**Figure 8: Left**: Accuracy-fairness curves of baseline random forest models v.s. fair models on the ENEM dataset. **Right**: Quantiles of per-sample score std. across high/low fairness models and baseline.

those individuals for which std. increases from 0 to a large positive value (median around > 0.15). Hence, the blue area shows that significant arbitrariness is introduced by the fairness intervention, in addition to and separate from the effects of thresholding the baseline.

### E.4 Complete Experimental Plots

In order to evaluate the predictive multiplicity of models with similar accuracy and fairness levels, we divide the accuracy-fairness frontier plots into 8x8 grids and put models in the corresponding bins. To compare the arbitrariness of models satisfying high/low fairness constraints, we select bins in two different MEO ranges and bins with baseline models. Then, we compute the standard deviation of scores of models corrected by REDUCTION in the three bins (high fairness/low fairness/baseline) and plot the quantile curves. We use PANDAS package's quantile function with its default linear interpolation method.

Across three baseline model classes (random forest, gradient boosting, and logistic regression), fair models exhibit higher score arbitrariness. Especially at top quantiles, all fair models have standard deviations in scores going up to 0.5. This means that for the individuals at the top percentile, the model prediction can flip if another random seed is used in model training.

Furthermore, we evaluate the predictive multiplicity of models corrected by different fairness intervention methods. Across datasets, all fairness intervention methods exhibit maximal standard deviation of scores of 0.5 at top quantiles for random forest baseline methods. LEVERAGING [12] exhibit score arbitrariness comparable to that of baseline for GBM and Logistic Regression methods. REJECTION and LEVERAGING output thresholded scores directly, while REDUCTION outputs probabilities (with most scores close to 0 or 1).

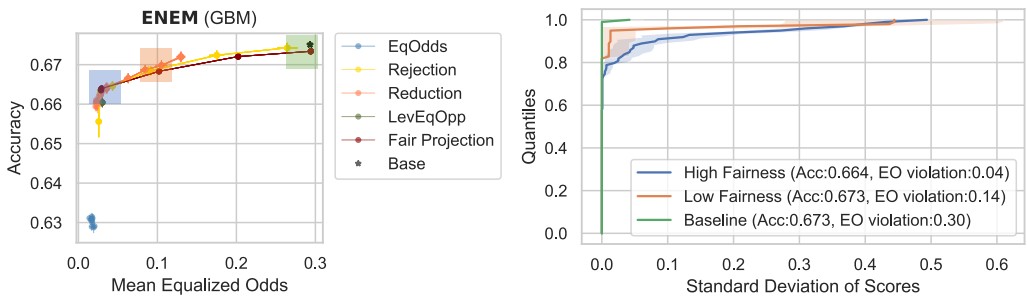

**Figure 9: Left**: Accuracy-fairness curves of baseline gradient boosting models (GBM) v.s. fair models on the ENEM dataset. **Right**: Quantiles of per-sample score std. across high/low fairness models and baseline.

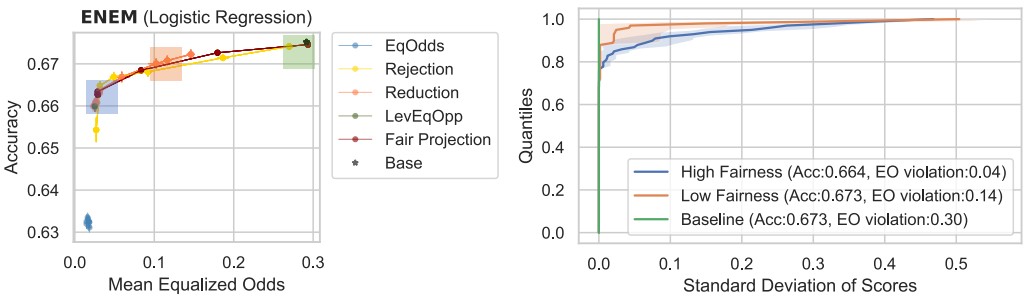

**Figure 10: Left**: Accuracy-fairness curves of baseline logistic regression models v.s. fair models on the ENEM dataset. **Right**: Quantiles of per-sample score std. across high/low fairness models and baseline.

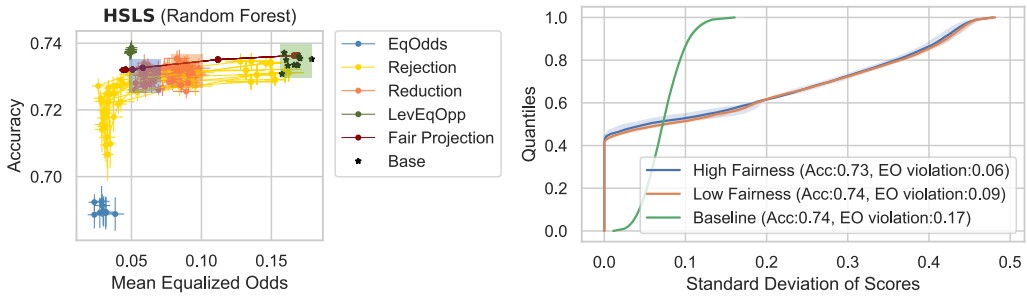

**Figure 11: Left**: Accuracy-fairness curves of baseline random forest models v.s. fair models on the HSLS dataset. **Right**: Quantiles of per-sample score std. across high/low fairness models and baseline.

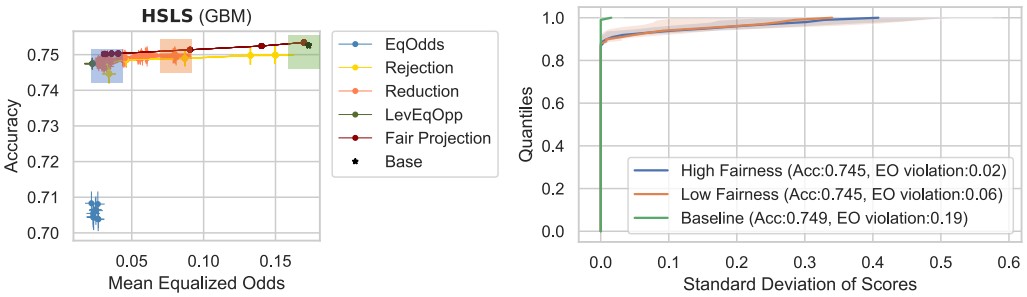

**Figure 12: Left**: Accuracy-fairness curves of baseline gradient boosting models (GBM) v.s. fair models on the HSLS dataset. **Right**: Quantiles of per-sample score std. across high/low fairness models and baseline.

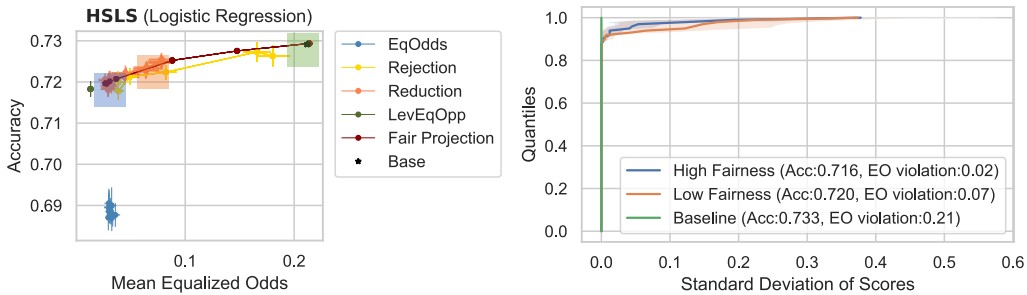

**Figure 13: Left**: Accuracy-fairness curves of baseline logistic regression models v.s. fair models on the HSLS dataset. **Right**: Quantiles of per-sample score std. across high/low fairness models and baseline.

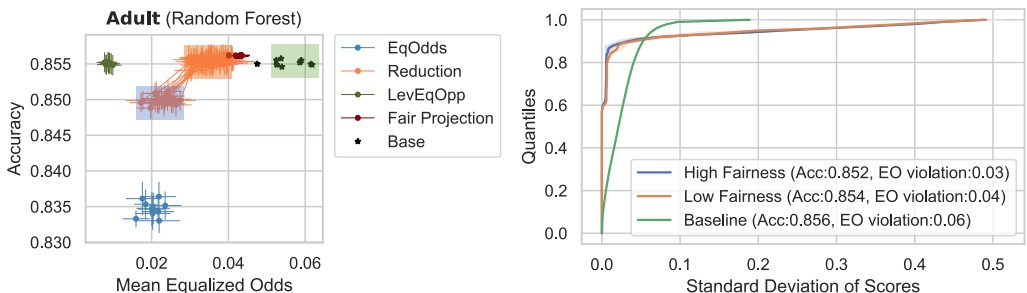

**Figure 14: Left**: Accuracy-fairness curves of baseline random forest models v.s. fair models on the Adult dataset. **Right**: Quantiles of per-sample score std. across high/low fairness models and baseline.

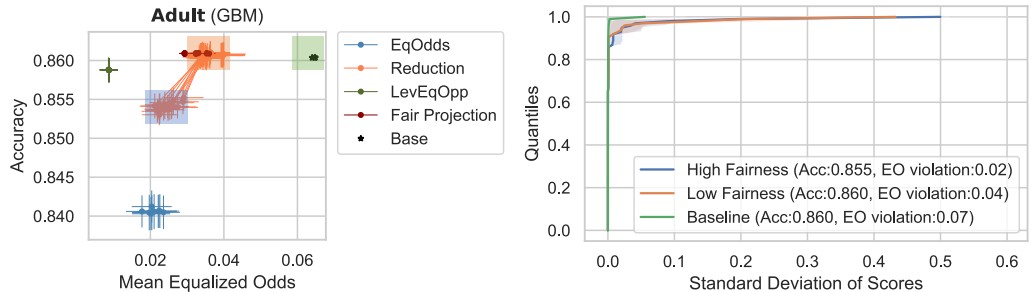

**Figure 15: Left**: Accuracy-fairness curves of baseline gradient boosting models (GBM) v.s. fair models on the Adult dataset. **Right**: Quantiles of per-sample score std. across high/low fairness models and baseline.

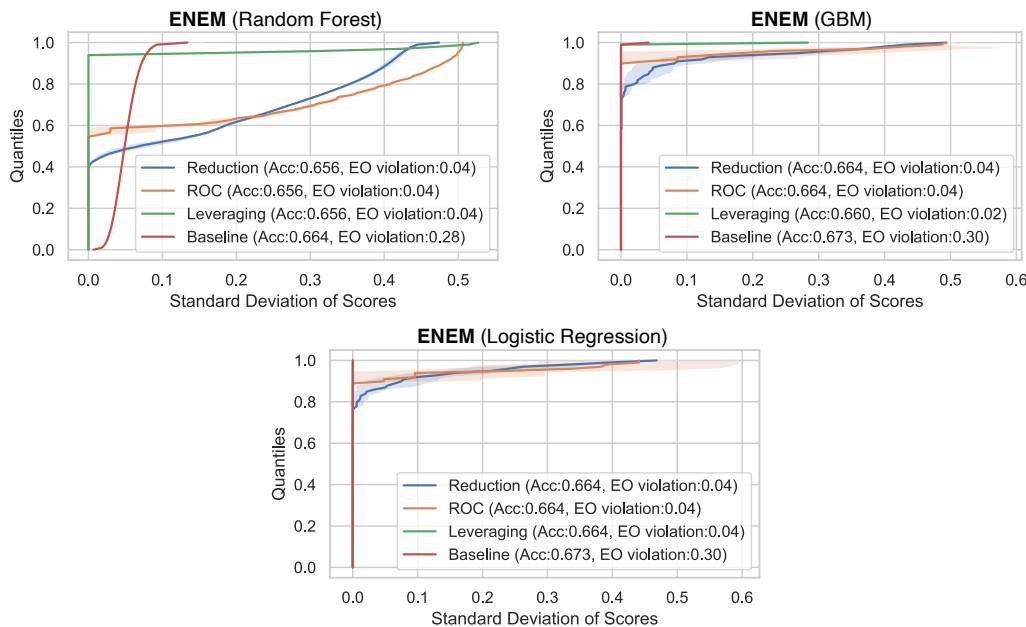

**Figure 16:** Quantile plot on models in high-fairness bin for various fairness interventions v.s. baseline models on ENEM. Fair models produce larger score std. at top percentiles compared to the baseline model.

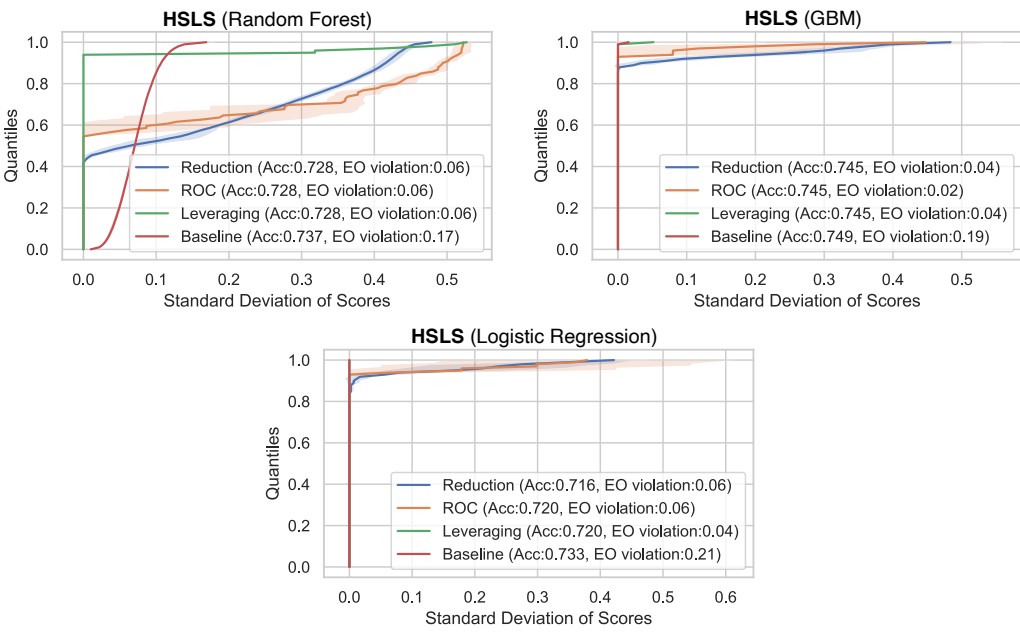

**Figure 17:** Quantile plot on models in high-fairness bin for various fairness interventions v.s. baseline models on HSLS. Fair models produce larger score std. at top percentiles compared to the baseline model.

