# OpenReview forum: "Individual Arbitrariness and Group Fairness"
_NeurIPS.cc/2023/Conference — NeurIPS 2023 spotlight_

### Official Review · Reviewer_gGVC · 2023-06-16

**Soundness:** 3 good
**Presentation:** 4 excellent
**Contribution:** 4 excellent
**Rating:** 7
**Confidence:** 3

**Summary:**

This paper explores the interaction of predictive multiplicity with fairness interventions, making the observation that models which receive common fairness interventions often result in increased predictive multiplicity within the Rashomon set. They back this up with some theoretical exploration and experiments, proposing ensembling as an approach for reducing multiplicity.

**Strengths:**

- the observation that models under fairness interventions have higher predictive multiplicity is a novel and interesting one
- the paper is well-presented and well-scoped, making a clear and useful contribution
- Figure 3 is a nice intuitive explanation for why the observed phenomenon occurs
- ensembles are a reasonable mitigation for this problem

**Weaknesses:**

- I'm a little confused by the discussion around "confidence" in 4.2. It would be useful to give more explanation of how confident (or unconfident) classifiers improve the multiplicity problem, and if this matters before or after ensembling. I also cannot seem to find the referred discussion around fairness and confidence in the Appendix - it sounds like an interesting observation and would be interesting to highlight it a bit more.
- some questions around the formalisms in Sec 2: 1. I think T(D) should be a distribution rather than a set, 2. the writing of equation 1 is a little confusing - it's not quite clear how both m and epsilon can be parameters (what if there aren't m models of loss less than epsilon? is it the m lowest loss or any m?) 3. is the domain of \ell above Eq 1 [0, 1] x {0, 1}?
- in Prop 3.1, is there some assumption made around what loss function is used?


**Questions:**

- is reducing predictive multiplicity ultimately a good thing? the upside is that the output is more consistent, but the downside is that the more consistent output is not necessarily chosen in a principled manner. I'd be interested to hear more discussion of this tradeoff
- I'm curious about why the Leveraging approach improves multiplicity so much - this would be good to explore a bit if you have space or at least discuss
- do we know that ensembles will satisfy fairness metrics if their component models do?

---

> ### Author Rebuttal · Authors · 2023-08-10
>
> We thank Reviewer gGVC for the thoughtful comments and for appreciating the novelty and value of the work. We hope that the answers below address the points raised in the review.
>
> ---
>
> - **Q1: “Is reducing predictive multiplicity ultimately a good thing?”**
>
>     --- We thank the Reviewer for this important question! There are mainly two lines of thought on this topic. One line is motivated by the growing recognition that at least some forms of arbitrariness are harmful (common response). For a discussion on positive views of predictive multiplicity, see Kulynych et al. [30], Section 7.1.
>
>     When predictive multiplicity poses harm to individual users, we believe having a mechanism that stabilizes training is important. One danger of multiplicity is when the model developer is unaware that trivial and innocuous choices during training (even the choice of random seed for parameter initialization!) may lead to arbitrary predictions during model deployment. Our research aims to spotlight this issue, revealing that the addition of group fairness constraints does not necessarily preclude arbitrariness (see **Fig. R1** and our **global response** in this rebuttal). Our proposed solution—an ensembling method—retrains the model with different hyperparameters to produce multiple models that satisfy both fairness and accuracy constraints. As shown in Fig. 6, this approach not only leads to more consistent predictions but also maintains baseline fairness values.
>     We will incorporate this discussion, including the trade-off you've highlighted, into the concluding section of our paper.
>
>
> ---
>
> - **Q2:** **“why the Leveraging approach improves multiplicity so much”**
>
>     ---  The primary goal of the leveraging approach is to identify an optimal threshold that reduces the disparity of opportunities based on distinct groups while ensuring optimal accuracy. This threshold is applied to generate the final decision output, i.e., binary outcomes in a classification task, as opposed to outputting the likelihood of class memberships.
>
>     In cases where the variance of scores is low, implementing thresholding on the scores can yield a more stable and consistent output, except for those points near the decision boundary. The leveraging approach, much like other group-fairness-oriented interventions such as the rejection approach, focuses primarily on decision (i.e., binary) outputs rather than raw scores (i.e., values in $[0,1]$). These fairness interventions essentially result in a compression of score variance, contributing to the straight line at 0 in **Fig. R3**.
>
>     We must note, however, that not all fairness interventions exhibit this pattern. For instance, calibration-based interventions, which are predicated on score outputs, may yield results quite distinct from the leveraging approach, which outputs either 0 or 1. This is because, while group-fairness notions focus on predictions, calibration is based on scores.
>
>     To further clarify this point, we have supplemented the rebuttal with Figure R3, which demonstrates the effect of thresholding the scores of the baseline model. The behavior exhibited by this curve exhibit resemblance with that of the leveraging approach, but, as shown in the violin plot and discussed in the global response, the increase in score std dev and the ensuing arbitrariness persists after thresholding.
>
>
> ---
>
> - **Q3:** **“do we know that ensembles will satisfy fairness metrics if their component models do?”**
>
>     --- Great question! The answer depends on the nature of the fairness metrics in question.
>
>     If the group fairness metrics amount to convex constraints on model performance, ensembling — which is a linear combination of models — will still satisfy the fairness constraints. For example, score-based formulations of Statistical Parity, Equalized Odds, and Overall Accuracy Equality are examples of fairness metrics that can be expressed as convex (in fact, linear) constraints on probabilistic classifiers that output scores instead of thresholded predictions; see Alghamdi et al. [2, Appendix A.7] for a thorough discussion.
>
>     However, when fairness metrics are cast in terms of thresholded classifier outputs (e.g., differences in FPR and FNR across groups), we cannot provide a theoretical guarantee that fairness constraints will continue to hold post-ensembling due to the metrics’ non-convex nature. Nevertheless, we observe empirically that these fairness constraints are typically met by the ensemble model, as demonstrated in **Figure 6**. We will stress this limitation in the revised version of the paper.
>
>
> ---
>
> - **Q4: “It would be useful to give more explanation of how confident (or unconfident) classifiers improve the multiplicity problem, and if this matters before or after ensembling.”**
>
>     --- We thank the Reviewer for raising this question! In general, a classifier giving very confident scores, say 0.95 for class 1 or 0.05 for class 0, does not improve the multiplicity problem. While the score is often interpreted as “confidence”, the score is sometimes a proxy for the label, and has no implication of the classifier’s confidence. For example, in Fig.1 of [20], when given a picture of a dog, all scores assigned by competing models are high, but on different classes.
>
>     We add the confidence assumption, before ensembling, to prove the concentration of predictions in ensembled models. We say a classifier is confident when, roughly, its scores are bounded away from 0.5 with high probability. This assumption enables us to translate consistent scores among ensembled models to consistent predictions in Theorem 4.2.
>
>
> ---
>
> - **Q5: “some questions around the formalisms in Sec 2”**
>
>     --- We appreciate the Reviewer for pointing out details where the notation can be improved! We will make the relevant changes to the notation of the empirical Rashomon set in Section 2.

---

> > ### Comment · Reviewer_gGVC · 2023-08-11
> > **Response**
> >
> > Thanks for the rebuttal - I already have a strong score on this paper so no need to update anything on my end.

---

### Official Review · Reviewer_APJf · 2023-07-04

**Soundness:** 3 good
**Presentation:** 2 fair
**Contribution:** 4 excellent
**Rating:** 7
**Confidence:** 3

**Summary:**

The authors discuss arbitrariness in automated decision-making, i.e. the fact that similarly trained classifiers may end up disagreeing on the most suitable label for a data point. In particular, the authors observe that arbitrariness can be worsened by fairness interventions, suggesting that increased group fairness may mean that some individuals arbitrarily receive worse decisions. A solution is proposed in the form of an ensemble algorithm that takes many (arbitrarily) fair classifiers and takes a majority vote.

**Strengths:**

The authors visit the active yet understudied topic of arbitrariness in a seemingly novel way. Not only may fairness metrics be oblivious to arbitrariness, their optimization may even worsen it. This nuance meaningfully adds to the debate.

The arguments are presented in a simple manner, and the theoretical results are only as complicated as they need to be.

The paper is clearly thought-provoking and it invites discussion at a level fit for NeurIPS.

**Weaknesses:**

1. My biggest concern for the paper is a philosophical one: why is arbitrariness (under your definition) harmful?
    * In Example 1, this is not very clear: in the 'fair' column, each group receives mistakes at the same rates. Why is arbitrariness problematic here?
    * I have a similar concern with Example 2: it is elegantly shown that two classifiers are each equally EO-fair yet use a different threshold (leading to arbitrariness). Why should individuals care which classifier is chosen?
    * Perhaps the argument could be made that arbitrariness is a form of unfairness, and that certain fairness notions are oblivious to it. Yet also in this case I wonder whether there is not always a fairness notion that *does* appropriately consider arbitrariness. For example in Figure 2, the arbitrariness could be detected through the Equalised Opportunity fairness notion (i.e. equal false negative rates). In that case, model 2 would be considered fair and model 1 would not be.
    * Please note I have not read up on the socio-technical arguments for studying arbitrariness. Nevertheless, I believe the examples in this paper should clarify why arbitrariness is important.

2. Figure 4 presents a worrying result. Should we not be able to see the orange curve (low fairness) trend towards the green curve (baseline)? To their credit, the authors remark this as a remarkable result, but I wonder whether the authors can discuss why we don't see a more gradual shape evolution from green to orange as the fairness intervention is done more and more intensively. If not, then it raises the question if the baseline curve is comparable to the others.

3. Many parts of the presentation are unpolished. Here are some examples.
    * Equations 1, 2 and 3 all use different symbols for the score / classifier function. Equation 2 in particular is inconsistent, as it indexes the data samples with $i$, and it selects models as elements from a set whereas other equations use the symbol $i$.
    * It seems $\mu$ in Eq. 3 should also have a subscript $x$.
    * Definition 3 presents the random variable $\hat{Y}$ whereas $h(x)$ was used before.
    * Section 4 rather suddenly makes the jump from binary classification to multiclass classification.
    * Figure 3 unnecessarily has a different font and layout. Its x-axis is unlabeled on the right, even though it seems different from the x-axis on the left. The colors in Figure 5 should also be more consistent (as the same methods are used on both sides).

4. Something appears to be wrong with the LaTeX settings. There are no line numbers, and the "Anonymous Authors" block is hidden. If this was done to save space, then this gives an unfair advantage compared to other authors that do follow the template correctly.

**Questions:**

Please find my questions and suggestions in the Weaknesses box.

If the authors have limited time during the rebuttal, then please consider only answering my questions related to the first weakness.

**Limitations:**

The (negative) societal impact of the work was not discussed. The authors may want to consider whether ensemble classifiers are a reasonable solution to arbitrariness in practice. Fairness in general has many societal risks if implemented incorrectly.

---

> ### Author Rebuttal · Authors · 2023-08-10
>
> We thank Reviewer APJf for the thoughtful comments and for appreciating the novelty and value of the work. We hope that the answers below address the points raised in the review.
>
> ---
>
> - **Q1: ‘’My biggest concern for the paper is a philosophical one: why is arbitrariness (under your definition) harmful?’’**
>
> --- We thank the Reviewer for raising this important question! Please refer to the **global response** in this rebuttal for an in-depth discussion of arguments and examples presented in the literature as to why arbitrariness is harmful. In particular, we include under the paragraph **"Why is arbitrariness (i.e., predictive multiplicity) harmful?"** in our response a discussion that will be reflected in the final version of the paper. We hope this response clarifies your concerns.
>
> The Reviewer also raised an important point on whether some fairness notions capture arbitrariness while others are oblivious to it. We would like to first note that Figure 2 is a simplified example. In general, harmful arbitrariness is a **global** notion, one that can only be measured with a pool of equally good models, not with a single classifier. In contrast, group-fairness notions are **local**: they pertain to a *single* classifier and its average performance across groups. Hence, harmful arbitrariness cannot *a priori* be captured by *local* notions such as group fairness. Our work shows that this disentanglement between fairness and arbitrariness is extreme: one can be orthogonal to the other (as can be inferred by, e.g., our Proposition 3.1; see also our generalization result included in the response to **Reviewer 47CU**'s **Q1**).
>
> ---
>
> - **Q2: “Figure 4 presents a worrying result. Should we not be able to see the orange curve (low fairness) trend towards the green curve (baseline) (…) as the fairness intervention is done more and more intensively?”**
>
> --- Thanks for the great question! It is indeed natural to expect that as a fairness intervention is applied with a less-stringent group fairness constraint, the scores’ standard deviation tend towards the baseline. However, this is not what we observe, and this phenomena actually reveals the inherent overconfidence, almost thresholded-like, scores from the fairness intervention. Increased arbitrariness still exist even comparing the fair classifiers with the thresholded baseline models. Please refer to our detailed discussion under the paragraph **"Explanation of Attached Figures to the Rebuttal"** in the global response.
>
> To be clear, the classifiers chosen in Figure 1 and 4—obtained from the reductions approach using the implementation from the IBM AIF360 package—outputs scores. However, the majority of the samples receive 0 or 1 post-intervention (see Figure R4). Once we threshold the baseline scores and compute the relevant statistics, the thresholded baseline curve and those with fairness intervention have a closer resemblance (see Figure R3)—thresholding produces the initial phase transition at 0 and high standard deviation at top quantiles.
>
> Beyond the samples that have 0 standard deviation in scores (i.e., consistent prediction) in the thresholded baseline model to begin with, the result remains worrying. As shown in Figure R2, removing the samples that receive consistent predictions both pre and post intervention (about 40% of them), there is a notable group (blue region), the biggest group by count, where the standard deviation increased significantly post-intervention.
>
> ---
>
> - **Q3: ‘’Many parts of the presentation are unpolished. Here are some examples. (...)’’**
>
> --- We appreciate the Reviewer for pointing out details and parts of the paper where the notation and layout can be clarified and improved. We will make all the suggested changes in the revised version of the paper.
>
> 1. Use $h$ to denote classifiers in equations (2) and (3);
> 2. Use appropriate indices in equation (2);
> 3. Clarify that $\bar{\mu}$ denotes the empirical mean for a fixed $x$;
> 4. Change the random variable $\hat{Y}$ to $h(X)$ in equation (5);
> 5. Use the notation of binary classification in Section 4 and explain in Appendix the generalization to multi-class classification;
> 6. Clean up Figure 3 and 5 for consistency in font, layout, and color.
>
> Thanks a lot for providing the detailed feedback that further strengthens the paper!
>
> ---
>
> - **Q4: ‘’Something appears to be wrong with the LaTeX settings.’’**
>
> --- This is an unfortunate and accidental mistake because of an oversight on our side: we used the “preprint” instead of the “review” option in the NeurIPS style file. We recompiled the document under the “preprint” option and observed no meaningful change in length. We regret the error.
>
> ---
>
> - **Q5: ‘’The (negative) societal impact of the work was not discussed. The authors may want to consider whether ensemble classifiers are a reasonable solution to arbitrariness in practice. Fairness in general has many societal risks if implemented incorrectly.’’**
>
> --- We thank the Reviewer for the valuable remark. Fairness interventions are at risk of creating the illusion of fairness if not implemented correctly—we will add this note to the limitation section. Ensembling acts as a stabilizing step in mitigating arbitrariness—rather than stopping at a single fair and accurate model, we retrain the model several times with different hyperparameters (e.g. random initialization) to get many models that satisfy both fairness and accuracy constraints. As shown in Fig. 6, this strategy reduces score variation while maintaining baseline fairness and accuracy values. For large models, where training is costly, such an ensembling procedure may be computationally expensive. It is definitely valuable to explore a more computationally efficient strategy in such cases. We will add a discussion in the limitation section as well.

---

> > ### Comment · Reviewer_APJf · 2023-08-12
> >
> > I'm convinced by the author's response to my biggest concern, which was the lack of motivation for arbitrariness. I especially appreciate the distinction that arbitrariness is a global notion affecting multiple classifiers. Therefore, I increase my score and trust the authors will implement the promised changes.
> >
> >  It is clear the paper will be accepted, and I congratulate the authors for their excellent submission.

---

### Official Review · Reviewer_MNFT · 2023-07-05

**Soundness:** 4 excellent
**Presentation:** 3 good
**Contribution:** 4 excellent
**Rating:** 8
**Confidence:** 4

**Summary:**

The authors theoretically and empirically study predictive multiplicity as it relates to fairness and accuracy. They show in both ways that multiplicity increases as a result of bias mitigation.

**Strengths:**

This is a problem that has not been studied and was worth studying because practitioners don't think about it, and should start doing so.

Both the theory and the experiments are compelling.

The paper is well written.

The math looks correct.

Bravo.

**Weaknesses:**

The authors should justify why arbitrariness is so bad under limited resources that can be allocated. If two people are exactly the same and right at the decision boundary, what other than an arbitrary decision is just if there is only one spot? Similarly, the authors should mention that several bias mitigation post-processing algorithms including [23] have an arbitrariness nature to them as part of the way they work.

It would be nice if the authors could connect their concentration results more to the original work in (non-fairness) ensemble classifiers, including Breiman's original paper on Random Forests (https://link.springer.com/article/10.1023/a:1010933404324) and the parameters used therein: strength and correlation. There may be simple ways of doing so by extending the operating points of  https://doi.org/10.1109/TKDE.2012.219 to also consider fairness metrics.



**Questions:**

The paper is pretty clear.

**Limitations:**

Nothing specific noted.

---

> ### Author Rebuttal · Authors · 2023-08-10
>
> We thank Reviewer MNFT for the thoughtful review and for appreciating the novelty, complexity, significance, and positive prospects of the work!
>
> ---
>
> **Q1: “Justify why arbitrariness is so bad under limited resources. If two people are exactly the same and right at the decision boundary, what other than an arbitrary decision is just if there is only one spot?”**
>
> --- This is a great point. We agree that, within the context of a single model, making an arbitrary decision between two individuals who are indistinguishable in scores (i.e., equally-distant to the boundary) seems fair. However, when viewed from the perspective of a collection of equally effective models, one person may end up being more likely to experience a change in prediction than the other ************across************ models, resulting in a disparate incidence of arbitrariness.
>
> In your example, equal distance to a boundary for a single fixed model does not necessarily imply equal likelihood of score change across a collection of competing models. While conventional fairness measures address the average performance of a single classifier across groups, arbitrariness (as considered in our paper) and predictive multiplicity is a global concept that requires a suite of equally accurate and fair models in order to be measured.
>
> The potentially detrimental impact of disparate arbitrariness has been increasingly recognized in recent literature [12,14,28]. Although arbitrariness might be unavoidable—especially when dealing with over-parametrized models such as neural networks—serious harm can occur when (i) the selection of a model from a set of "equally good and fair" options disproportionately impacts certain individuals, (ii) the model is employed in high-stakes applications with significant individual consequences, and/or (iii) a single model is used across multiple institutions, possibly leading to systemic exclusion for certain individuals (the “Algorithmic Leviathan” in [12]). We delve deeper into this topic in our global response — please take a look.
>
> A key point of concern arises when the arbitrariness in model selection remains unknown or unacknowledged. In many fairness interventions, like existing implementations of the “Reductions Approach,” the output for a sample is a thresholded score, in which case a decision (0-1 output) is always given for an individual output. In your example, an apparently inconsequential choice, such as the random seed used to initialize model parameters or a fairness intervention, can dictate the selected individual, transforming a seemingly innocuous decision into a potentially harmful one. Of greater concern is when this harm is silent, and model selection is oblivious to such disparate arbitrariness and masked by the assurance of group fairness. We hope our work sheds light on this issue.
>
> ---
>
> **Q2: “Mention that several bias mitigation post-processing algorithms including [23] have an arbitrariness nature to them as part of the way they work.”**
>
> --- Thanks for the Reviewer’s suggestion. Indeed, several bias-mitigation algorithms used to ensure group fairness have inherent randomness that may exacerbate arbitrariness, and we will highlight this issue. In fact, even outside of fairness, Kulynych et al. [26] show that Differential Privacy mechanisms could also unintentionally increase arbitrariness by adding randomness to ensure privacy.
>
> This phenomenon also seems intertwined with the behavior of fairness interventions outputting overly confident classifiers; please see **Fig. R4** and the accompanying discussion in our common reply in this rebuttal.
>
> ---
>
> **Q3: “Connect their concentration results more to the original work in (non-fairness) ensemble classifiers”**
>
> --- Thanks for the Reviewer’s suggestion! It would be an interesting future direction to extend the existing work on ensemble classifiers and specifically Random Forests to consider fairness metrics. (We note that the referenced concentration results pertain to the probability of error, whereas our concentration results have to do with score agreement between ensembled classifiers.)

---

> > ### Comment · Reviewer_MNFT · 2023-08-10
> >
> > Thanks for the rebuttal. I think the paper should be accepted as long as the authors do include their additional discussion of why arbitrariness is harmful.

---

### Official Review · Reviewer_47CU · 2023-07-19

**Soundness:** 4 excellent
**Presentation:** 4 excellent
**Contribution:** 4 excellent
**Rating:** 7
**Confidence:** 4

**Summary:**

This paper studies the effect of group fairness regularization constraints on a notion termed arbitrariness. Here, arbitrariness is defined to be the variability of the predictions, for a set of similarly accurate models, on individual samples. In this paper, they show that when a model is regularized to satisfy group fairness constraints, it is still possible that the predictions of similarly accurate models show variability in the individual predictions. The set of similarly accurate models on a given subpopulation of data is termed the rashomon set. Here, the authors measure arbitrariness with the quantile of the standard deviation of the output of the models in a rashomon set. They also measure ambiguity by computing the proportion of points in the dataset that can be assigned multiple conflicting predictions. They prove that it is possible for a model to satisfy accuracy equality (i.e. all protected subgroups have similar accuracy), but for the arbitrariness of the rashomon set to be as high as up to 100 percent. To counter this challenge, they propose a simple scheme that ensembles the models in the rashomon set (convex combination), and show theoretically and empirically (on three datasets for random forest models) that this ensembling scheme reduces arbitrariness.

**Strengths:**

**Originality**\
This paper follows in the line of work on predictive multiplicity and underspecification. Overall, this paper explores the interplay between the predictive multiplicity of a rashomon set and and group fairness properties of that set. A key insight in this work that I hadn't seen before is that it is possible for group fairness constraints to increase the predictive multiplicity of the rashomon set. This finding seems counter-intuitive. The ensembling approach discussed in this work has been mentioned in past work, like the underspecification paper by D'amour et. al., however, it was not explicitly used to address predictive multiplicity as it is used here. Overall, this paper furthers the discussion in a particularly interesting way.

**Quality/Clarity**\
The paper is well-written and clear. Figure 2 was quite helpful for trying to understand the main message of the paper. Overall, this is a high quality paper.

**Significance**\
If it remains the case that arbitrariness is orthogonal to group fairness constraints, then that is an important finding and has far reaching impact on how to obtain models with reliable predictions. The paper has opened up a line of work that can be explored in several ways.

**Weaknesses:**

None of the weaknesses I discuss here are disqualifying, but I note below places where the paper can be improved.

**Unify the terms Predictive multiplicity, Arbitrariness, and Ambiguity**: Right now, I think these three terms are all referring to the same general phenomenon, but it is unclear whether there is an instantiation of each that is different. Perhaps the authors could unify this.

**Proposition 3.1 is only for accuracy parity/equality**: Unclear if this proposition generalizes to other metrics (see questions section).

**Restricted model class and dataset**: This paper only explores random forest models on tabular datasets.

**Questions:**

- Is equality of accuracy interchangeable with any other group fairness constraint in proposition 3.1? I know Example 2 discusses equality of opportunity, but the main argument for why group fairness constraints and predictive multiplicity are perhaps orthogonal is this proposition, but in reading the proof, it is not clear to me that you can extend this to other fairness metrics. Accuracy parity makes sense here, but I don't think something like group calibration fits the theme. Can the authors comment on this?


- How does one pick the number of models in a rashomon set? In practice, in the experiments, it seems like you need to set a number for the size of the rashomon set, and this parameter might have an oversized influence on the arbitrariness and other properties of the set. How was this number determined?

**Limitations:**

Discussed in the final section of the paper.

---

> ### Author Rebuttal · Authors · 2023-08-10
>
> We thank Reviewer 47CU for the thoughtful comments and for appreciating the novelty and value of the work. We hope that the answers below address the points raised in the review.
>
> ---
>
> **Q1: Proposition 3.1 shows orthogonality of group fairness and predictive multiplicity. Can you extend equality of accuracy to other group fairness constraint (e.g. group calibration) in this proposition?**
>
> --- Thanks for the question! We confirm that the orthogonality result in Proposition 3.1 extends to group-fairness constraints beyond Overall Accuracy Equality (OAE), which we will add to the Appendix in the final version. What changes now is that there will always be a floor for the probability of error (denoted $\\epsilon_{\\min}$), and then the condition for attaining maximal ambiguity would be to have at least $m > 1/(\\epsilon - \\epsilon_{\\min})$ competing models. (Note that $\\epsilon_{\min}=0$ for OAE!)
>
> We illustrate the case of Statistical Parity (SP) below. We note that group calibration can also be done in the more general setup of probabilistic prediction, so the probability of error is taken with respect to the population distribution. We will add further details on this generalized setup for group calibration in the final version of the paper.
>
> Consider SP for binary classes and groups: $\\mathrm{Pr}(\\hat{Y} = 1 \\mid S=0) = \\mathrm{Pr}(\\hat{Y} = 1 \\mid S=1)$. If $Y$ does not satisfy SP, then any predictor $\\hat{Y}$ satisfying SP must make an error of at least
>
> $$ \\epsilon_{\\min} := P_Y(1) - \\min(P_{Y|S=0}(1) , P_{Y|S=1}(1)).$$
>
> We have the following analogue of Proposition 3.1.
>
> **Proposition 3.1 (For Statistical Parity).** For small $\\epsilon > \\epsilon_{\\min}$, and $m$ large enough, there is an empirical Rashomon set $\\hat{\\mathcal{R}}_m=\\{ h_1, \\cdots , h_m\\}$ such that: 1) each classifier $h_i$ has probability of error $\\epsilon$; 2) each classifier $h_i$ satisfies Statistical Parity (SP) exactly; and 3) the set $\\hat{\\mathcal{R}}_m$ has $100\\%$ ambiguity.
>
>  In particular, under mild assumptions on $P_{Y,S}$, it suffices to have  $m> 1/(\\epsilon-\\epsilon_{\\min})$ models.
>
> **Proof.** Denote $p=P_{Y|S=0}(1), q=P_{Y|S=1}(1)$. Assume $p \\le q$.
>
> Let $\\alpha_{0,0}=\\epsilon-\\epsilon_{\\min}$, $\\alpha_{1,0}=1-\\alpha_{0,0}$, $\\alpha_{0,1} = \\frac{1-p}{1-q}\\alpha_{0,0}$, and $\\alpha_{1,1}=\\frac{p}{q} \\alpha_{1,0}$. For each $(y,s)\\in \\{0,1\\}^2$, let $h$ be a classifier that assigns the class $1$ to a fraction $\\alpha_{y,s}$ of the individuals $x$ that have true class $y$ and group $s$. (We have $0<\\alpha_{y,s}<1$ if, e.g.,  $\\epsilon < \\epsilon_{\\min} + (1-q)/(1-p)$.) The classifier $h$:
>
> 1. has probability of error $\\epsilon$;
> 2. and satisfies SP exactly.
>
> By partitioning the sets of individuals into subsets of relative sizes $r_{y,s}:=\\min(\\alpha_{y,s},1-\\alpha_{y,s})$, we may construct $m>1/\\min_{y,s}(r_{y,s})$ classifiers as above that make conflicting predictions on each individual (similar to the proof of Proposition 3.1), i.e., the constructed set has ambiguity $100\\%$. Finally, if, e.g., $q \\le 1-p(1-p)$ (and $\\epsilon$ is correspondingly small enough), then the condition simplifies into $m> 1/(\\epsilon-\\epsilon_{\\min})$.
>
> ---
>
> **Q2: “How does one pick the number of models in a rashomon set? How was this number determined in the experiments?”**
>
> --- We thank the Reviewer for the question. The number of models sampled from the Rashomon Set indeed may have an important influence on the arbitrariness metric being measured, but the importance varies with the chosen metric. For example, ambiguity (percentage of samples receiving a conflicting label) is non-decreasing with the number of models. On the other hand, standard deviation of scores—our metric of choice—is less influenced by the choice of the hyper-parameters. In practice, the number of models one can sample depends heavily on available computation resources. Through literature research and some initial experimentation, we find 10 models being a reasonable number to estimate the standard deviation of scores, given the size of the datasets, computation required to train the 3 baseline model classes with the various fairness interventions. Repeating the result using 10 random splits of the data also produce small error bars (in shaded region), which acts as a sanity check that the estimation is fairly consistent. There is a separate question of how many models one should sample in order to accurately estimate a metric of arbitrariness within a Rashomon set. This pertains to a line of work that investigates the size of a Rashomon set, which also depends on the volume of the hypothesis space. We refer the Reviewer to Hsu and Calmon [20] for a related discussion.
>
> ---
>
> **Q3: “Unify the terms Predictive multiplicity, Arbitrariness, and Ambiguity.”**
>
> --- Thanks for the suggestion! The three terms are connected in the following way. Arbitrariness refers to the general phenomena where a choice — a model or a decision — cannot be justified. Arbitrariness can result from predictive multiplicity, which pertains to prediction tasks and refers to the phenomena where multiple competing models yield conflicting predictions. Ambiguity, proposed by Marx et al. [28], is also a metric of predictive multiplicity of a dataset.
>
> We understand the confusion since we sometimes use these terms informally. We will add the above discussion to clarify the terminology in the revised version.
>
> ---
>
> **Q4: “Restricted model class and dataset: This paper only explores random forest models on tabular datasets.”**
>
> --- Thanks for the comment! We would like to point out that we have provided results for two other model classes—gradient boosting and logistic regression—in **Section D** of the Appendix, as discussed in **Section 5** of main body.

---

> > ### Comment · Reviewer_47CU · 2023-08-11
> > **Acknowledging Rebuttal**
> >
> > Thanks for response to clear the issues that I had. Overall, I think this work is a nice contribution, so I'll maintain my current rating.

---

### Author Rebuttal · Authors · 2023-08-10

We thank all reviewers for their time and effort! We are glad our paper was positively received. In particular, we were encouraged that all reviewers recognized the novelty and impact of our work: “If it remains the case that arbitrariness is orthogonal to group fairness constraints, then that is an important finding and has far-reaching impact on how to obtain models with reliable predictions. The paper has opened up a line of work that can be explored in several ways.” (**Reviewer 47CU)**; “This is a problem that has not been studied and was worth studying because practitioners don't think about it, and should start doing so.” (**Reviewer MNFT**); “The authors visit the active yet understudied topic of arbitrariness in a seemingly novel way” (**Reviewer APJf**); and “the observation that models under fairness interventions have higher predictive multiplicity is a novel and interesting one” (**Reviewer gGVC**).

We appreciate the reviewers’ thoughtful input. We believe we have addressed all critical points in the rebuttal below and have outlined how we plan to update the paper accordingly.

---

### **Why is arbitrariness (i.e., predictive multiplicity) harmful?**

In response to **Reviewer MNFT Q1, Reviewer APJf Q1, Reviewer gGVC Q1**, we provide a further discussion (will be added to the revision) to contextualize this work. Our work is motivated by recent literature that delineates the harmful impact of arbitrariness. See Creel and Hellman [12] for a philosophical overview of the hazards of arbitrariness in algorithmic decision-making, the introduction of Marx et. al [28] for potential harms of predictive multiplicity in ML, and D’Amour et. al [14] for extensive examples on how underspecification and the ensuing arbitrary choices among competing models can challenge the credibility of ML models.

There is a growing recognition that at least some types of arbitrariness in ML can be viewed as harmful; and in particular *disparate* arbitrariness. Arbitrariness itself may be inevitable— when models are overparametrized. Harm arises when (i) an arbitrary choice of model across the set of “equally good and fair” models impacts different individuals differently, (ii) the model is deployed in a high-stakes application with individual-level consequences, and/or (iii) the same model is used across institutions, where an arbitrary choice of model may lead to systemic exclusion for certain individuals. One example from [12, 28] is:

- **Recidivism prediction:** consider a recidivism risk prediction task that admits competing models with similar accuracy and group fairness levels. In this case, a person who is predicted to recidivate by one model may achieve a lower recidivism risk score by a competing model. An arbitrary choice between these models may lead to unjustified harm, e.g., some defendants may receive consistent predictions across competing models whereas others may not. The harm is particularly concerning when the existence of competing models is unknown or not communicated to the model user. In such cases, arbitrariness can strengthen the growing calls to forgo the use of ML in criminal justice.

As thoroughly discussed by Creel and Hellman [12] (“Algorithmic Leviathans”), arbitrariness may lead to systemic exclusion of opportunities for certain individuals when a single decision-making system, out of the many equally good ones, is chosen without deliberation.

---

### **Explanation of Attached Figures to the Rebuttal**

We would like to direct your attention to the attached page, with further analysis the fairness-intervention-induced arbitrariness. These figures aim to address points raised by **Reviewer APJf (Q2) and Reviewer gGVC (Q2 and Q4)**. In short, these figures illustrate three points: **1)** Fairness intervention makes classifiers overly confident (attached **Fig. R4**); **2)** This overconfidence partially explains the increase in arbitrariness after intervention; and **3)** Additional significant arbitrariness is induced post-intervention that affects a sizable portion of the population that had no arbitrariness pre-intervention (attached **Fig. R2**).

- In response to **Reviewer APJf**’s question on the missing trend in **Fig. 4** in the manuscript, in **Fig. R3**, we added 3 curves to the original **Fig. 1** in the main paper, including different levels of intervention using the Rejection approach [23] as well as the thresholded baseline. From the similarity in the shape of the thresholded baseline curve and the fair models’ curves, thresholding-like behavior of some interventions may explain some—but certainly not all (see **Fig. R2**)—increase in score std dev and the ensuing arbitrariness. We plot in **Fig. R2** the distribution of score std. relative to the *thresholded* baseline model. Removing samples that receive very low score std. both from thresholded baseline and fair classifiers, the largest group (blue area) in this violin plot are those individuals for which std. increases from 0 to a large positive value (median around > 0.15). Hence, the blue area shows that significant arbitrariness is introduced by the fairness intervention, in addition to and separate from the effects of thresholding the baseline. Combining the findings from **Figs. R2** and **R3** reveals that overconfidence of scores post-intervention partially explains the shape of the cumulative plots in the original **Fig. 1** of the manuscript.
- Lastly, the aforementioned overconfidence of the post-intervention classifiers is illustrated in **Fig. R4** (addressing **Reviewer gGVC**’s **Q4**). **Fig. R4** provides a histogram of the scores of fair classifiers (from the Reductions [1] approach), which displays the inherent overconfidence, almost thresholded-like, scores from post-intervention.

---

Please do follow up with us if you have additional suggestions and feedback that can further strengthen the paper. Thank you very much!

---

### Decision · Program_Chairs · 2023-09-21

**Decision:**

Accept (spotlight)

**Comment:**

The paper explores the phenomenon of predictive multiplicity where similar performing models can produce very different predictions on individual samples. The paper demonstrates that procedures for statistical based fairness metrics often exacerbate the predictive multiplicity. The paper argues that so-called arbitrariness should be considered in addition to usual accuracy and fairness and proposes an ensemble algorithm agnostic to the fairness intervention that can ensure more stable predictions with theoretical guarantees.

The reviewers all enjoyed reading the paper and enthusiastically recommended acceptance.

In addition, there are a couple of related works that may be relevant about “churn” which appears to have a very similar definition to predictive multiplicity:

[1] D. Bahri, and H. Jiang. "Locally adaptive label smoothing for predictive churn." ICML 2021

[2] Bhojanapalli, S., Wilber, K., Veit, A., Rawat, A. S., Kim, S., Menon, A., & Kumar, S. (2021). On the reproducibility of neural network predictions. arXiv preprint arXiv:2102.03349